

# Large deviations in the symmetric simple exclusion process with slow boundaries: A hydrodynamic perspective

**Soumyabrata Saha⋆ and Tridib Sadhu†**

Department of Theoretical Physics, Tata Institute of Fundamental Research,
Homi Bhabha Road, Mumbai 400005, India

⋆ soumyabrata.saha@tifr.res.in , † tridib@theory.tifr.res.in

## Abstract

We revisit the one-dimensional model of the symmetric simple exclusion process *slowly* coupled with two unequal reservoirs at the boundaries. In its non-equilibrium stationary state, the large deviations functions of density and current have been recently derived using exact microscopic analysis by Derrida, Hirschberg and Sadhu in *J Stat Phys* **182, 15 (2021)**. We present an independent derivation using the hydrodynamic approach of the macroscopic fluctuation theory (MFT). The slow coupling introduces additional boundary terms in the MFT-Action, which modifies the spatial boundary conditions for the associated variational problem. For the density large deviations, we explicitly solve the corresponding Euler-Lagrange equations using a simple *local* transformation of the optimal fields. For the current large deviations, our solution is obtained using the additivity principle. In addition to recovering the expression of the large deviations functions, our solution describes the most probable path for these rare fluctuations.

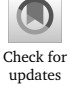

# 1   Introduction

Naturally occurring processes typically operate outside of equilibrium. Their examples abound, ranging from living matter, atmospheric activities to mechanically driven systems. Out-of-equilibrium processes exhibit a wide array of unique features, sometimes even surprising in comparison to equilibrium systems, making them interesting while also presenting new challenges for theoretical analysis. Characterising macroscopic properties of non-equilibrium systems in ways akin to equilibrium statistical mechanics is an exciting endeavour for modern Physics.

In the pursuit of a widely applicable theoretical approach, a prevalent idea is to characterize non-equilibrium fluctuations in terms of large deviations function (ldf) [1,2]. Analogous to the equilibrium free energy function, the ldf describes macroscopic properties. For instance, generic long-range correlation in non-equilibrium stationary state is reflected in the non-locality of ldf [1,3–5], and phase transitions relate to singularity of ldf [6–11]. However, calculating ldf for many-body non-equilibrium systems is challenging, even numerically [12–16], and often theoretical calculations rely on the tools of integrability [1,17,18] that are specific to curated models, which are otherwise few and far between.

A remarkable development came in the early 2000s from the seminal work of Bertini, De Sole, Gabrielli, Jona-Lasinio, and Landim [19]. They proposed a fluctuating hydrodynamics approach to calculate ldf for interacting many-body systems with diffusion dominated dynamics. This influential approach, popularly known as Macroscopic Fluctuation Theory (MFT) [19,20], has been successfully applied in a wide range of non-equilibrium scenarios, including transport models [3,5,21–24], non-equilibrium phase transitions [6–8,25–27], interface fluctuations [28,29], single-file diffusion [23,30,31]. Related ideas have found applications in weak turbulence [32] and oceanography [33]. A notable recent development is the relation of the MFT to classical integrability [18,29,34–36]. A detailed account of the works on MFT can be found in the extended review article [37] and a recent shorter review [38].

In this work, we revisit the applications of MFT to the exclusion processes [1,17], where the theory has been particularly successful. The early triumph of MFT was in reproducing the ldf of density [19,39–41] and current [20,42] in the *symmetric simple exclusion processes* (SSEP). These exact results were derived by Derrida and collaborators using solution of the microscopic dynamics [1, 43–47]. In these examples, the SSEP is in the *non-equilibrium stationary state* (NESS), driven by coupling with two boundary reservoirs of unequal densities, where the boundary rates are of the same order as in the bulk. The exact results of the ldf for both density and current have been recently extended [48] for the case where the boundary rates are slower. These extensions confirm a desirable robustness of the fluctuations in the NESS, similar to equilibrium states. Following [48], we shall refer to the former boundary coupling as 'fast coupling' and the latter as 'slow coupling'. The significance of slow boundaries has been emphasized in a long list of early works [49–56], which emphasized how the Dirichlet boundary conditions for hydrodynamic density transform into Robin boundary conditions in the presence of a slow bond. Many variants of lattice gas with slow rates have been studied in [57–62]. Early studies on exclusion processes can be found in the Physics literature [43–46, 63–71] and in the Mathematics literature [72–75] with emphasis on applications in transport systems [17,76,77].

Reproducing the slowly coupled extensions of ldf using the framework of MFT poses considerable challenges. Firstly, the individual bonds linking the system and the reservoirs become influential, and one might naively expect that the large-scale theory of MFT would not capture such single-bond effect. Secondly, extending the Dirichlet boundary condition for density and the response fields in MFT, which were set assuming instantaneous equilibration at the boundary for fast coupling, is not straightforward for slow coupling. Thirdly, earlier methods for the variational solution of MFT are not suitable for the modified boundary conditions in the slow coupling regime. In this work, we overcome these challenges and extend MFT for beyond the fast coupling regime, recovering the exact results of ldf for the slow-boundaries SSEP from [48].

In presenting this work, we adopt a novice approach by reconstructing some of the well-established fundamental steps from a perspective that we believe would be accessible to a general Physics reader. These steps include a derivation of the hydrodynamic equation for the average density with a Robin boundary condition for slowly coupled SSEP and a derivation of the MFT-Action, incorporating contributions from slow boundaries. To focus our discussion at the new results, we have relegated these derivations to the Appendix. In the main text, we begin by recalling the exact results of ldf for the slow-boundaries SSEP from [48]. These results are summarized in Sec. 2 alongside our new results for the fluctuating hydrodynamics in the context of slowly coupled boundaries. In the subsequent Sec. 3, we introduce the variational formalism of MFT for the ldf of density of SSEP with slow boundaries. We demonstrate that the corresponding Euler-Lagrange equation can be explicitly solved using a simple *local* transformation. Our new solution complements the earlier solution of the fast boundary case, obtained through ingenious non-local transformations [39, 40]. In our formulation, the fast boundary problem is merely a limiting case. Notably, our solution not only reproduces the exact expression [48] of the ldf of density for slow-boundaries SSEP but also presents the optimal path that leads to rare fluctuations, a feature absent in the microscopic solution [48]. Additionally, besides the Euler-Lagrange equation, we show that the ldf is a solution of the corresponding Hamilton-Jacobi equation. In Sec. 4, we tackle the corresponding variational problem for the ldf of current for slow-boundaries SSEP, determining the optimal path and reproducing the exact result for the ldf [48]. We conclude in section 5 with an outlook on the potential generalization for a larger class of diffusive systems.

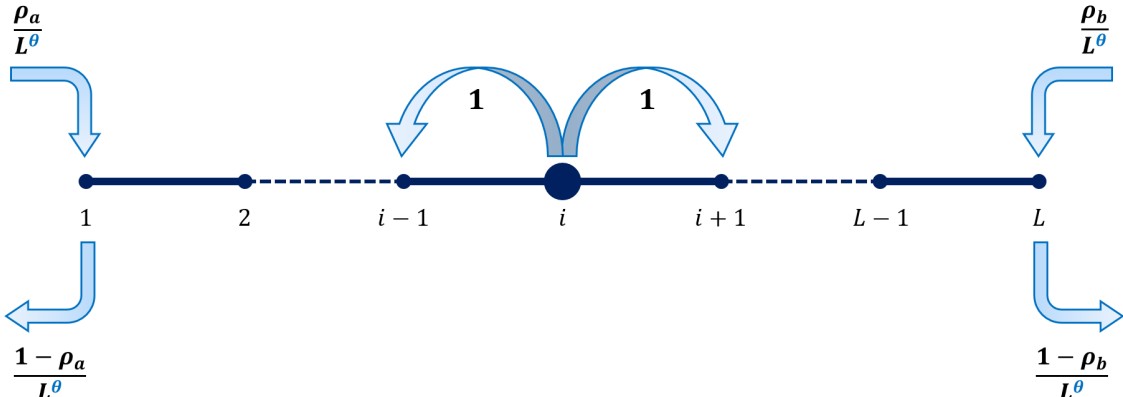

Figure 1: Jump rates of particles in the SSEP on a linear chain of $L$ sites coupled with reservoirs of density $\rho_a$ on the left and $\rho_b$ on the right. The parameter $\theta$ controls the time scale of jumps between the system and the reservoirs.

## 2 A summary of the old and the new results

The SSEP is a lattice-gas model defined (see Fig. 1) on a finite one-dimensional lattice of $L$ sites indexed by $i \equiv \{1, 2, \ldots, L\}$. At a particular instance, a lattice site can at most be occupied by one particle. Due to this simple exclusion, the configuration space is specified by the set of binary occupation variables $n_i$ which takes value 0 or 1 depending whether the $i$-th site is empty or occupied. On the $2^L$ configuration space the system evolves following a continuous-time Markov dynamics where a particle can jump across any of the bonds inside the lattice in either direction with unit rate provided that the target site is empty. In addition, if the boundary site $i = 1(L)$ is empty then a particle is deposited at rate $\rho_{a(b)}/L^\theta$ and if it is occupied then the particle is evaporated at rate $\left(1 - \rho_{a(b)}\right)/L^\theta$, where $\theta$ is a real-valued parameter. The rates emulate [1] coupling of the boundary sites $i = 1(L)$ to reservoirs of density $\rho_{a(b)}$. The parameter $\theta$ controls the time scale for jumps between the system and the reservoirs. For earlier considerations of similar parameters see [49, 54, 56].

### 2.1 Large deviations of density

At long times, the system reaches a non-equilibrium stationary state with a non-trivial probability of occupation variables represented by a Matrix product ansatz [1, 78]. For large $L$, the probability $P[\rho(x)]$ of coarse-grained density $n_i \simeq \rho\left(\frac{i}{L}\right)$ has a large deviation asymptotic that depends on the coupling parameter $\theta$. We recall the result from [48].

- Fast coupling regime: for $\theta < 1$ the jump rates across the boundary are faster compared to the relaxation rate ($\sim 1/L$) of density fluctuations inside the bulk.[1] This means for bulk density fluctuations the boundary sites are effectively equilibrated at the reservoir densities. For large $L$, the probability

$$\Pr[\rho(x)] \sim e^{-L\psi_{\text{fast}}[\rho(x)]}, \tag{1a}$$

where the ldf $\psi_{\text{fast}}[\rho(x)]$ is the function obtained in [39, 44] for $\theta = 0$.

$$\psi_{\text{fast}}[\rho(x)] = \int_0^1 dx \left[ \rho(x) \log \frac{\rho(x)}{F(x)} + \left(1 - \rho(x)\right) \log \frac{1 - \rho(x)}{1 - F(x)} + \log \frac{F'(x)}{\rho_b - \rho_a} \right], \tag{1b}$$

---

[1] A hand waiving way to see this is from $\partial_t \sum_{i=2}^{L-1} \langle n_i \rangle = \left(\langle n_L \rangle - \langle n_{L-1} \rangle\right) - \left(\langle n_2 \rangle - \langle n_1 \rangle\right) \sim 1/L$.

is a non-local function of $\rho(x)$ where $F(x)$ is a monotone function related to $\rho(x)$ by the nonlinear differential equation

$$\rho(x) = F(x) + \frac{F(x)\big(1 - F(x)\big) F''(x)}{F'(x)^2}, \tag{1c}$$

with the Dirichlet boundary condition $F(0) = \rho_a$ and $F(1) = \rho_b$.

- Slow coupling regime: for $\theta > 1$ at large $L$, the bulk is effectively in equilibrium at an average density whose fluctuations are governed by the boundary rates. For large $L$, the probability

$$\Pr[\rho(x)] \sim e^{-L \psi_{\text{slow}}[\rho(x)]}, \tag{2a}$$

where

$$\psi_{\text{slow}}[\rho(x)] = \int_0^1 dx \left[ f(\rho(x)) - f(\varrho) \right] \quad \text{with } f(r) = r \log r + (1 - r) \log(1 - r), \tag{2b}$$

is the ldf [1,79] for a SSEP in equilibrium with a fixed integrated density $\varrho = \int_0^1 dx\, \rho(x)$. Fluctuations in $\varrho$ are suppressed following probability

$$P(\varrho) \sim e^{-L^\theta \phi(\varrho)} \quad \text{with} \quad \phi(\varrho) = \log \frac{4(\rho_a - \varrho)(\varrho - \rho_b)}{(\rho_a - \rho_b)^2}. \tag{2c}$$

The minimum of $\phi(\varrho)$ is at $\bar{\varrho} = \frac{1}{2}(\rho_a + \rho_b)$ where $\phi(\bar{\varrho})$ vanishes. The result (2) is derived using the exact correspondence between correlations of occupation variables for $\theta > 0$ and $\theta = 0$ discussed in [48]. The value of $\bar{\varrho}$ is in accord with the the exact microscopic result [79] for the steady state density

$$\langle n_i \rangle \equiv \bar{\rho}\left( \frac{i}{L} = x \right) \simeq \bar{\varrho} + \frac{1}{L^{\theta - 1}} [\rho_a(1 - x) + \rho_b x - \bar{\varrho}] \quad \text{with } \bar{\varrho} = \frac{\rho_a + \rho_b}{2}, \tag{3}$$

in the slow coupling regime and independently confirmed by the hydrodynamics in (A.9b).

- Marginal boundary coupling: the $\theta = 1$ is the marginal case between fast coupling and slow coupling regimes. In fact, fluctuations in both regimes can be obtained from appropriate limits of the marginal case. To see this, it is useful to introduce parameters $\Gamma_a$ and $\Gamma_b$ in the boundary rates as shown in Fig. 2.

For large $L$, the probability of density has the large deviation asymptotic [48]

$$\Pr[\rho(x)] \sim e^{-L \psi[\rho(x)]}, \tag{4a}$$

where

$$\psi[\rho(x)] = \int_0^1 dx \left[ \rho(x) \log \frac{\rho(x)}{F(x)} + \big(1 - \rho(x)\big) \log \frac{1 - \rho(x)}{1 - F(x)} + \log \frac{F'(x)}{\rho_b - \rho_a} \right]$$
$$+ \Gamma_a \log \frac{F(0) - \rho_a}{\Gamma_a(\rho_b - \rho_a)} + \Gamma_b \log \frac{\rho_b - F(1)}{\Gamma_b(\rho_b - \rho_a)}, \tag{4b}$$

with $F(x)$ being a monotonic solution of the differential equation in (1c), but now satisfying Robin boundary condition

$$F(0) - \Gamma_a F'(0) = \rho_a \quad \text{and} \quad F(1) + \Gamma_b F'(1) = \rho_b. \tag{4c}$$

It is readily seen that the ldf (1b) for fast coupling is the $\Gamma_{a(b)} \to 0$ limit of (4b). Similarly, $\Gamma_{a(b)} \to \infty$ limit of (4b) gives the slow coupling result (2b), which can be seen by expanding the former in powers of small $\Gamma_{a(b)}$. An analysis of this limit is presented in the Appendix I.

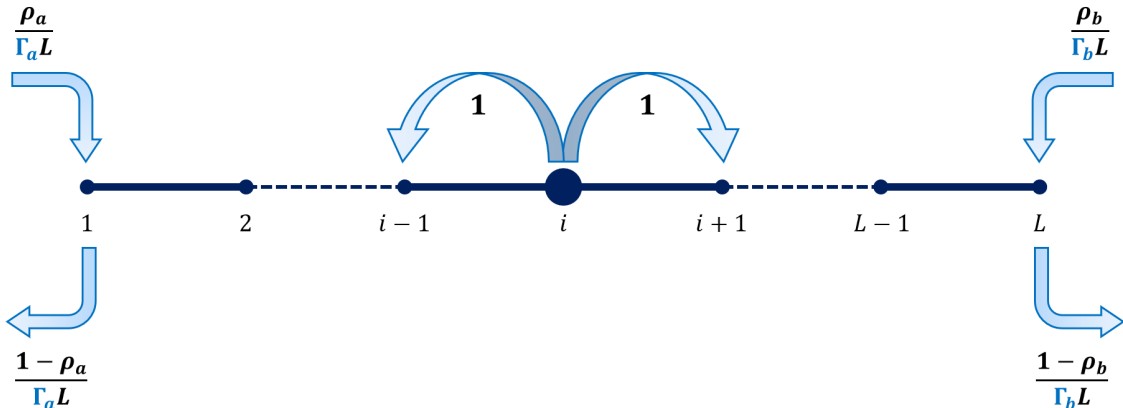

Figure 2: Jump rates for a SSEP with marginal coupling with reservoirs at the boundaries. Compared to Fig. 1 the differences are in the boundary rates.

## 2.2 Large deviations of current

Beside the density, another relevant quantity which has been extensively studied for SSEP is the empirical current $q = \frac{Q_{\mathcal{T}}}{\mathcal{T}}$ where $Q_{\mathcal{T}}$ is the net flux of particles across the system in a total duration $\mathcal{T}$. For large $\mathcal{T}$, the distribution of $q$ follows a large deviation asymptotic

$$\Pr(q) \sim e^{-\mathcal{T}\phi_L(q)}, \tag{5}$$

where the ldf $\phi_L(q)$ relates to the corresponding scaled cumulant generating function (scgf)

$$\mu_L(\lambda) = \lim_{\mathcal{T}\to\infty} \frac{1}{\mathcal{T}} \log \langle \exp(\lambda Q_{\mathcal{T}}) \rangle, \tag{6}$$

by a Legendre transformation $\phi_L(q) = \max_\lambda \left(\lambda q - \mu_L(\lambda)\right)$. For the SSEP, the dependence of the scgf on the parameters $\lambda$, $\rho_a$, and $\rho_b$ comes through a function [45,46]

$$\omega(\lambda, \rho_a, \rho_b) = \left(e^\lambda - 1\right)\rho_a(1-\rho_b) + \left(e^{-\lambda} - 1\right)\rho_b(1-\rho_a), \tag{7}$$

which is related to a symmetry of the tilted generator [22,46,80,81]. We recall the results of the scgf from [48].

In the large $L$ limit, the scgf $\mu_L(\lambda)$ has different asymptotics depending on the coupling parameter $\theta$.

- Fast coupling regime: for $\theta < 1$, the current fluctuation is dominated by transport inside the bulk. The cumulant generating function is identical to the $\theta = 0$ case [45,46] and is given by

$$\mu_L(\lambda) \simeq \frac{1}{L} R_{\text{fast}}\left(\omega(\lambda, \rho_a, \rho_b)\right), \quad \text{with} \quad R_{\text{fast}}(\omega) = \left(\text{arcsinh}\sqrt{\omega}\right)^2. \tag{8}$$

- Slow coupling regime: for $\theta > 1$, the two bonds connecting the system and the reservoirs are the bottleneck. The scgf is dominated by the transport across these two bonds and for large $L$,

$$\mu_L(\lambda) \simeq \frac{1}{L^\theta} R_{\text{slow}}\left(\omega(\lambda, \rho_a, \rho_b)\right), \tag{9a}$$

with

$$R_{\text{slow}}\left(\omega(\lambda, \rho_a, \rho_b)\right) = \max_\rho \min_{\widehat{\rho}} \left[\omega(\widehat{\rho}, \rho_a, \rho) + \omega(\lambda - \widehat{\rho}, \rho, \rho_b)\right], \tag{9b}$$

obtained following the additivity argument in [48] and using the scgf across a single bond. The optimization simplifies using eqn. (29) of [48]

$$R_{\text{slow}}(\omega) = \min_z \left[ \sinh^2 z + \sinh^2 \left( \sinh^{-1} \sqrt{\omega} - z \right) \right] = -1 + \sqrt{\omega + 1}. \tag{9c}$$

- Marginal boundary coupling: the $\theta = 1$ is the marginal case between the fast and slow coupling regimes, where the rates are defined in Fig. 2. For large $L$, the scgf [48]

$$\mu_L(\lambda) \simeq \frac{1}{L} R\big(\omega(\lambda, \rho_a, \rho_b)\big), \tag{10a}$$

with $R(\omega)$ written in a variational form

$$R(\omega) = \min_{z_a, z_b} \left[ \left( \sinh^{-1} \sqrt{\omega} - z_a - z_b \right)^2 + \frac{\sinh^2 z_a}{\Gamma_a} + \frac{\sinh^2 z_b}{\Gamma_b} \right]. \tag{10b}$$

The $R_{\text{fast}}(\omega)$ in (8) is the $\Gamma_{a(b)} \to 0$ limit of (10b). In the slow coupling limit $\Gamma_{a(b)} \to \infty$, $R(\omega) \simeq 0$ which is consistent with the scaling with $L$ in (9a).

## 2.3 Hydrodynamics of SSEP with the marginal boundary coupling

Considering that the fast and the slow coupling regimes are suitable limits of the marginal coupling we shall confine our discussion only to the latter, for which the jump rates are shown in Fig. 2. The current work is about recovering the expression of the ldf of density in (4b) and of current in (10a) using the hydrodynamic approach of MFT.

Hydrodynamics for SSEP describes evolution of the coarse-grained density $\rho(x, t) \simeq n_i(\tau)$ with $(x, t) \equiv \left( \frac{i}{L}, \frac{\tau}{L^2} \right)$ for large $L$, where $n_i(\tau)$ is the occupation variable at site $i$ at a microscopic time $\tau$. For the marginal boundary coupling, it has been rigorously shown [49,51,54] that the average density profile $\bar{\rho}(x, t)$ evolves in time following a diffusion equation

$$\partial_t \bar{\rho}(x, t) = \partial_x^2 \bar{\rho}(x, t), \tag{11a}$$

with the Robin boundary condition

$$\bar{\rho}(0, t) - \Gamma_a \, \partial_x \bar{\rho}(0, t) = \rho_a \quad \text{and} \quad \bar{\rho}(1, t) + \Gamma_b \, \partial_x \bar{\rho}(1, t) = \rho_b. \tag{11b}$$

For $\Gamma_{a(b)} \to 0$, the boundary condition (11b) reduces to the usual Dirichlet condition [39,82] in the fast coupling limit, whereas for $\Gamma_{a(b)} \to \infty$, it reduces to a Neumann condition in the slow coupling limit. In the Appendix A we present a simple derivation of (11).

The density field $\rho(x, t)$ fluctuates around its average value $\bar{\rho}(x, t)$ and evolves following a stochastic differential equation

$$\partial_t \rho(x, t) = \partial_x^2 \rho(x, t) + \partial_x \eta(x, t), \tag{12a}$$

in the bulk $0 < x < 1$ with the boundary condition

$$-\partial_x \rho(x, t) = \eta(x, t) + \begin{cases} \xi_{\text{left}}(t) & \text{at } x = 0, \\ \xi_{\text{right}}(t) & \text{at } x = 1. \end{cases} \tag{12b} \tag{12c}$$

Here, $\eta(x, t)$ is a weak Gaussian noise with zero mean and covariance

$$\langle \eta(x, t) \eta(x', t') \rangle = \frac{1}{L} 2\rho(x, t)\big(1 - \rho(x, t)\big) \delta(x - x') \delta(t - t'), \tag{13a}$$

whereas the boundary noises $\xi_{\text{left}}(t)$ and $\xi_{\text{right}}(t)$ are delta-correlated in time with a moment-generating function

$$\left\langle e^{\int dt\,\lambda(t)\,\xi_{\text{left}}(t)} \right\rangle \sim \exp\left( \frac{1}{\Gamma_a} \int dt\,\omega\big(\lambda(t),\rho_a,\rho(0,t)\big) \right), \tag{13b}$$

$$\left\langle e^{\int dt\,\lambda(t)\,\xi_{\text{right}}(t)} \right\rangle \sim \exp\left( \frac{1}{\Gamma_b} \int dt\,\omega\big(\lambda(t),\rho(1,t),\rho_b\big) \right). \tag{13c}$$

The multiplicative noises in (12) are interpreted with Itô convention. Note that from (13b), $\langle \xi_{\text{left}}(t) \rangle = \frac{1}{\Gamma_a}\big(\rho_a - \rho(0,t)\big)$ and with this the left boundary condition in (11b) is recovered. Similar analysis applies for the right boundary.

Our derivation of the ldf (4) and the scgf (10) is essentially based on the fluctuating hydrodynamics equation (12-13). In the Appendix C we present a non-rigorous derivation of this fluctuating hydrodynamic equation. A similar derivation for SSEP on a semi-infinite geometry was presented in [83].

The fast coupling limit ($\Gamma_{a(b)} \to 0$) of (12-13) gives the well-known [38,40,41] fluctuating hydrodynamics with Dirichlet boundary conditions. For a relevant derivation we refer to [84].

## 3 Large deviations of density using the MFT

In the stationary state, the average density profile $\bar{\rho}(x)$ is the solution of Laplace's equation $\bar{\rho}''(x) = 0$ with the Robin boundary condition $\bar{\rho}(0) - \Gamma_a\,\bar{\rho}'(0) = \rho_a$ and $\bar{\rho}(1) + \Gamma_b\,\bar{\rho}'(1) = \rho_b$. The ldf (4b) characterizes the relative probability weight of density fluctuations around the profile $\bar{\rho}(x)$. This probability of a density profile $r(x)$ is [39–41,47] the sum of probability amplitudes of all evolutions that started far in the past ($t \to -\infty$) at $\bar{\rho}(x)$ and ended at $r(x)$ at the time of measurement ($t = 0$). In the hydrodynamic description (see Appendix B for a derivation) this transition probability is written as a path integral

$$\Pr[r(x)] = \int_{\bar{\rho}(x)}^{r(x)} [\mathcal{D}\widehat{\rho}][\mathcal{D}\rho] \exp\left\{ -L \int_{-\infty}^{0} dt \left[ \int_0^1 dx \left( \widehat{\rho}(x,t)\,\partial_t \rho(x,t) \right) \right.\right.$$
$$\left.\left. - H[\widehat{\rho}(x,t),\rho(x,t)] \right] \right\}, \tag{14a}$$

where $\widehat{\rho}$ is the Martin-Siggia-Rose-Janssen-De Dominicis (MSRJD) response field [85–88] and the effective Hamiltonian

$$H[\widehat{\rho}(x,t),\rho(x,t)] = H_{\text{bulk}}[\widehat{\rho}(x,t),\rho(x,t)] + \frac{1}{\Gamma_a} H_{\text{left}}[\widehat{\rho}(0,t),\rho(0,t)]$$
$$+ \frac{1}{\Gamma_b} H_{\text{right}}[\widehat{\rho}(1,t),\rho(1,t)], \tag{14b}$$

is made of contributions from the bulk and the two boundaries. The bulk term [37,39,41,47]

$$H_{\text{bulk}}[\widehat{\rho},\rho] = \int_0^1 dx \left[ \rho(x,t)\big(1-\rho(x,t)\big)\partial_x\widehat{\rho}(x,t) - \partial_x\rho(x,t) \right]\partial_x\widehat{\rho}(x,t), \tag{14c}$$

whereas the contribution from the left and the right boundary

$$H_{\text{left}}[\widehat{\rho},\rho] = \omega\big(\widehat{\rho}(0,t),\rho_a,\rho(0,t)\big), \text{ and, } H_{\text{right}}[\widehat{\rho},\rho] = \omega\big(\widehat{\rho}(0,t),\rho_b,\rho(0,t)\big). \tag{14d}$$

The boundary terms were recognized earlier in [39,41,80], however their contribution were subdominant in the fast coupling regime.

For a large system size $L$, the path integral in (14a) is dominated by the path of optimal Action leading to the large deviation asymptotic (4a) with the ldf

$$\psi[r(x)] = \int_{-\infty}^{0} dt \left[ \int_{0}^{1} dx \left( \widehat{\rho}(x,t) \, \partial_t \rho(x,t) \right) - H[\widehat{\rho}(x,t), \rho(x,t)] \right], \qquad (15)$$

optimized over all density and response fields with $\rho(x,-\infty) = \bar{\rho}(x)$ and $\rho(x,0) = r(x)$.

The optimal fields that minimise the Action satisfy the Euler-Lagrange equations

$$\partial_t \rho(x,t) - \partial_x^2 \rho(x,t) = -2 \, \partial_x \left[ \rho(x,t) \left( 1 - \rho(x,t) \right) \partial_x \widehat{\rho}(x,t) \right], \qquad (16a)$$

$$\partial_t \widehat{\rho}(x,t) + \partial_x^2 \widehat{\rho}(x,t) = -\left( 1 - 2\rho(x,t) \right) \left( \partial_x \widehat{\rho}(x,t) \right)^2, \qquad (16b)$$

in the bulk with the spatial boundary conditions

$$\Gamma_a \, \partial_x \widehat{\rho}(0,t) = \rho_a \left( e^{\widehat{\rho}(0,t)} - 1 \right) - \left( 1 - \rho_a \right) \left( e^{-\widehat{\rho}(0,t)} - 1 \right), \qquad (17a)$$

$$\Gamma_a \left[ 2\rho(0,t) \left( 1 - \rho(0,t) \right) \partial_x \widehat{\rho}(0,t) - \partial_x \rho(0,t) \right] = \rho_a \left( 1 - \rho(0,t) \right) e^{\widehat{\rho}(0,t)}$$
$$- \rho(0,t)(1-\rho_a) e^{-\widehat{\rho}(0,t)}, \qquad (17b)$$

at the left boundary and

$$\Gamma_b \, \partial_x \widehat{\rho}(1,t) = (1-\rho_b) \left( e^{-\widehat{\rho}(1,t)} - 1 \right) - \rho_b \left( e^{\widehat{\rho}(1,t)} - 1 \right), \qquad (18a)$$

$$\Gamma_b \left[ 2\rho(1,t) \left( 1 - \rho(1,t) \right) \partial_x \widehat{\rho}(1,t) - \partial_x \rho(1,t) \right] = \rho(1,t)(1-\rho_b) e^{-\widehat{\rho}(1,t)}$$
$$- \rho_b \left( 1 - \rho(1,t) \right) e^{\widehat{\rho}(1,t)}, \qquad (18b)$$

at the right boundary.

**Remark**: The spatial boundary conditions naturally emerge form the minimization in (15). In the fast coupling regime ($\Gamma_{a(b)} \to 0$) the boundary conditions reduce to Dirichlet condition $\rho(0,t) = \rho_a$, $\rho(1,t) = \rho_b$, $\widehat{\rho}(0,t) = 0$ and $\widehat{\rho}(1,t) = 0$ which were usually imposed prior to the minimization [1, 39, 40].

## 3.1 The optimal path

In order to solve for the optimal fields, we introduce a *local* transformation

$$\widehat{\rho}(x,t) = \log \frac{\rho(x,t) \left( 1 - F(x,t) \right)}{F(x,t) \left( 1 - \rho(x,t) \right)}, \qquad (19)$$

to make a change of variable from $\widehat{\rho}(x,t)$ to $F(x,t)$. The transformation is inspired by a relation that appeared in [47] while solving the Hamilton-Jacobi for the fast coupling limit and has appeared in related contexts [39, 40, 89].

In terms of the new variable $F(x,t)$, the Euler-Lagrange equation (16a) becomes

$$\partial_t \rho(x,t) + \partial_x^2 \rho(x,t) = 2 \, \partial_x \left[ \frac{\rho(x,t) \left( 1 - \rho(x,t) \right)}{F(x,t) \left( 1 - F(x,t) \right)} \partial_x F(x,t) \right], \qquad (20a)$$

while (16b) becomes

$$\partial_t F(x,t) + \partial_x^2 F(x,t) = 2 \left[ \partial_x^2 F(x,t) - \frac{\rho(x,t) - F(x,t)}{F(x,t) \left( 1 - F(x,t) \right)} \left( \partial_x F(x,t) \right)^2 \right]. \qquad (20b)$$

Remarkably, the spatial boundary conditions (17, 18) reduce to a simple Robin boundary condition for the $F$-field

$$F(0,t) - \Gamma_a \, \partial_x F(0,t) = \rho_a \,, \tag{21a}$$

$$F(1,t) + \Gamma_b \, \partial_x F(1,t) = \rho_b \,, \tag{21b}$$

and a similar condition for the density field

$$\rho(0,t) - \Gamma_a \, \partial_x \rho(0,t) = \rho_a + \frac{\big(1 - 2\rho(0,t)\big)\big(F(0,t) - \rho(0,t)\big)\big(F(0,t) - \rho_a\big)}{F(0,t)\big(1 - F(0,t)\big)} \,, \tag{22a}$$

$$\rho(1,t) + \Gamma_b \, \partial_x \rho(1,t) = \rho_b + \frac{\big(1 - 2\rho(1,t)\big)\big(F(1,t) - \rho(1,t)\big)\big(F(1,t) - \rho_b\big)}{F(1,t)\big(1 - F(1,t)\big)} \,. \tag{22b}$$

The optimal path is the solution of (20) with the spatial boundary conditions (21-22) and temporal conditions $\rho(x,-\infty) = \bar{\rho}(x)$ and $\rho(x,0) = r(x)$. The solution is given by the case where both sides of (20b) vanish such that the $F$-field satisfies an anti-diffusion equation

$$\partial_t F(x,t) + \partial_x^2 F(x,t) = 0 \,, \tag{23}$$

while the optimal density-field is expressed in terms of the optimal $F$-field at any time by the relation

$$\rho(x,t) = F(x,t) + \frac{F(x,t)\big(1 - F(x,t)\big)\partial_x^2 F(x,t)}{\big(\partial_x F(x,t)\big)^2} \,. \tag{24}$$

We have explicitly verified that this particular solution (23, 24) is consistent with (20a) (see Appendix D) and with the spatial boundary condition (22) (see Appendix E).

The solution is also consistent with the temporal condition $\rho(x,-\infty) = \bar{\rho}(x)$. This is seen from the fact that $\partial_x^2 \bar{\rho}(x) = 0$ with the Robin boundary condition (11b) and therefore $\bar{\rho}(x)$ is a fixed point of the anti-diffusion equation (23) with (21). This means there is a solution for $F(x,t)$ with $F(x,-\infty) = \bar{\rho}(x)$ for which the $\rho(x,t)$ in (24) is consistent with the condition $\rho(x,-\infty) = \bar{\rho}(x)$.

In summary, the optimal-Action path is given in terms of the solution of (23) with the boundary conditions (21), $F(x,-\infty) = \bar{\rho}(x)$, and $F(x,0)$ specified by (24) for $\rho(x,0) = r(x)$. The optimal density field $\rho(x,t)$ and the optimal response field $\hat{\rho}(x,t)$ are explicitly given in terms of $F(x,t)$ using (19, 24). Evolution of the optimal density $\rho(x,t)$ and the conjugate field $F(x,t)$ for specific parameter values are shown in Fig. 3.

**Remark**: Earlier solution [40, 41] of the Euler-Lagrange equation for the fast coupling limit was obtained using two step non-local transformations. Our solution using the local transformation (19) includes fast coupling as a limiting case.

## 3.2 The optimal Action

For the optimal Action in (15) we note that along the optimal path $H[\hat{\rho}, \rho] = 0$. This is obvious using the fact that Hamiltonian is conserved in Euler-Lagrange evolution and that the initial state at $t \to -\infty$ is a fixed point of the dynamics, where the Hamiltonian vanishes. We have also explicitly verified this in the Appendix F.

Substituting the vanishing $H$ and using (19) in (15) we get the ldf

$$\psi[r(x)] = \int_{-\infty}^{0} dt \int_{0}^{1} dx \left( \log \frac{\rho(x,t)\big(1 - F(x,t)\big)}{F(x,t)\big(1 - \rho(x,t)\big)} \, \partial_t \rho(x,t) \right). \tag{25}$$

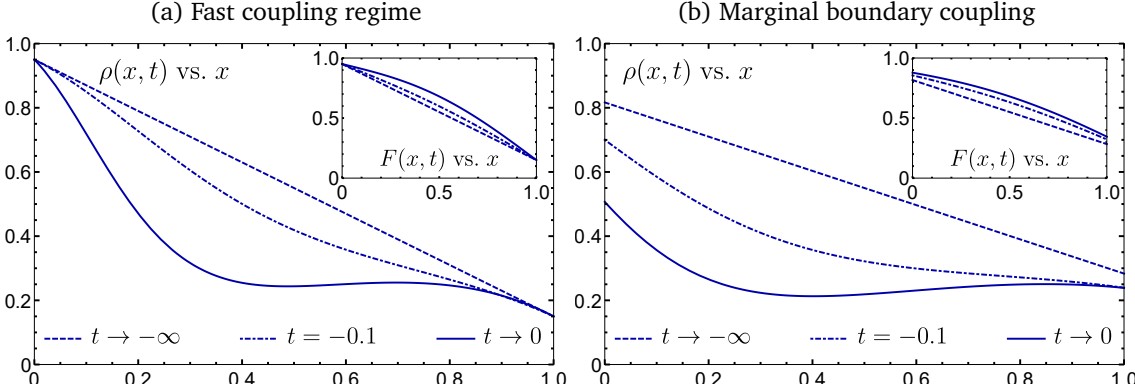

Figure 3: The optimal density profile $\rho(x,t)$ in (24) at different stages of evolution for the boundary reservoir densities $\rho_a = 0.95$ and $\rho_b = 0.15$. The figure (a) corresponds to the fast coupling regime with ($\Gamma_{a(b)} \to 0$) while the figure (b) corresponds to the marginal boundary coupling with $\Gamma_a = \Gamma_b = 0.25$. The inset shows the corresponding profile for the optimal field $F(x,t)$ in (23) at different times indicated in the legends.

The expression (25) explicitly depends on the entire optimal path, whereas the final result (4b) involves only the initial and the final states. The simplification comes from writing the integrand in (25) as a total time derivative. To show this, we start by using integration by parts in time-variable,

$$\psi[r(x)] = \int_{-\infty}^{0} dt \int_{0}^{1} dx \left\{ \partial_t \left[ \rho(x,t) \log \frac{\rho(x,t)\big(1-F(x,t)\big)}{F(x,t)\big(1-\rho(x,t)\big)} \right] - \frac{\partial_t \rho(x,t)}{1-\rho(x,t)} \right. $$
$$\left. + \frac{\rho(x,t)\,\partial_t F(x,t)}{F(x,t)\big(1-F(x,t)\big)} \right\} \tag{26}$$

Next, we use the following two identities. The first identity

$$-\frac{\partial_t \rho(x,t)}{1-\rho(x,t)} + \frac{\rho(x,t)\,\partial_t F(x,t)}{F(x,t)\big(1-F(x,t)\big)} = \partial_t\left( \log \frac{1-\rho(x,t)}{1-F(x,t)} \right) + \frac{\partial_t F(x,t)\,\partial_x^2 F(x,t)}{\big(\partial_x F(x,t)\big)^2}, \tag{27}$$

where we have used (24). The second identity

$$\frac{\partial_t F(x,t)\,\partial_x^2 F(x,t)}{\big(\partial_x F(x,t)\big)^2} = \partial_t\big( \log \partial_x F(x,t) \big) - \partial_x\left( \frac{\partial_t F(x,t)}{\partial_x F(x,t)} \right), \tag{28}$$

is a result of simple algebra. Substituting (27,28) in (26), we get

$$\psi[r(x)] = \int_{-\infty}^{0} dt \int_{0}^{1} dx \left\{ \partial_t \left[ \rho(x,t) \log \frac{\rho(x,t)}{F(x,t)} + \big(1-\rho(x,t)\big) \log \frac{1-\rho(x,t)}{1-F(x,t)} \right. \right.$$
$$\left. \left. + \log \partial_x F(x,t) \right] - \partial_x\left( \frac{\partial_t F(x,t)}{\partial_x F(x,t)} \right) \right\}. \tag{29}$$

Completing the integration over time for the term with total time derivative and integration over space for the term with space derivative, respectively and then using the spatial boundary

conditions (21), we get

$$
\psi\left[r(x)\right] = \int_0^1 dx \left[ r(x) \log \frac{r(x)}{F(x)} + \left(1 - r(x)\right) \log \frac{1 - r(x)}{1 - F(x)} + \log \frac{F'(x)}{\bar{\rho}'(x)} \right]
$$
$$
+ \Gamma_a \int_{-\infty}^0 dt \frac{\partial_t F(0,t)}{F(0,t) - \rho_a} - \Gamma_b \int_{-\infty}^0 dt \frac{\partial_t F(1,t)}{\rho_b - F(1,t)}, \tag{30}
$$

where, in addition, we used the temporal boundary conditions $\rho(x, -\infty) = F(x, -\infty) = \overline{\rho}(x)$ and $F(x, 0) = F(x)$ determined in terms of $\rho(x, 0) = r(x)$ from (24). The time integration in the last two terms in (30) involving the spatial boundary are simple and leads to

$$
\psi\left[r(x)\right] = \int_0^1 dx \left[ r(x) \log \frac{r(x)}{F(x)} + \left(1 - r(x)\right) \log \frac{1 - r(x)}{1 - F(x)} + \log \frac{F'(x)}{\bar{\rho}'(x)} \right]
$$
$$
+ \Gamma_a \log \frac{F(0) - \rho_a}{\bar{\rho}(0) - \rho_a} + \Gamma_b \log \frac{\rho_b - F(1)}{\rho_b - \bar{\rho}(1)}. \tag{31}
$$

As expected, the expression (31) involves only the density at the final and at the initial times. To express (31) in the form of the reported result (4b) we use [1,84]

$$
\bar{\rho}(x) = \rho_a \left( 1 - \frac{\Gamma_a + x}{1 + \Gamma_a + \Gamma_b} \right) + \rho_b \frac{\Gamma_a + x}{1 + \Gamma_a + \Gamma_b}, \tag{32}
$$

and absorb a constant term by setting $\psi\left[\bar{\rho}(x)\right] = 0$.

**Remark:** An equivalent way to verify that the optimal Action (15) indeed gives (4b) is by checking that the latter follows the Hamilton-Jacobi equation

$$
H\left[ \frac{\delta \psi\left[r(x)\right]}{\delta r(x)}, r(x) \right] = 0, \tag{33}
$$

with the Hamiltonian in (14b). For the fast coupling limit, corresponding equation was verified in [39,47] using a non-trivial transformation. In the Appendix G, we recall a derivation of (33) for the marginal boundary coupling and in Appendix H we explicitly verify the solution (4b).

## 4 Large deviations of current using the MFT

The MFT formulation of current fluctuation for SSEP in the fast coupling limit has been extensively reported in [20,37,80]. For a similar formulation with the marginal boundary coupling, we define the scaled cumulant generating function

$$
\mu(\{\lambda_i\}, \rho_a, \rho_b) \simeq \frac{1}{\mathcal{T}} \log \left\langle e^{\sum_{i=0}^L \lambda_i Q_i(\mathcal{T})} \right\rangle, \tag{34}
$$

for large $\mathcal{T}$, where $Q_i(\mathcal{T})$ is the net flow of particles from the $i$-th site to the $(i+1)$-th site in duration $\mathcal{T}$ with $Q_0$ being the flow from the left reservoir to the leftmost site ($i = 1$) and $Q_L$ being the flow from the rightmost ($i = L$) site to the right reservoir.

Following our reconstruction of the MFT for the marginal boundary coupling in Appendix B it is straightforward to show that, for $\mathcal{T} = L^2 T$ and a slowly varying function $\lambda_i \simeq \frac{1}{L} \alpha\left(\frac{1}{L}\right)$, the generating function in the large $L$ limit

$$
\left\langle e^{\sum_{i=0}^L \lambda_i Q_i(\mathcal{T})} \right\rangle \simeq \int [\mathcal{D}\widehat{\rho}][\mathcal{D}\rho] \exp \left\{ -L \int_0^T dt \left[ \int_0^1 dx \left( \widehat{\rho}(x,t) \partial_t \rho(x,t) \right) - H_{\text{bulk}}[h, \rho] \right. \right.
$$
$$
\left. \left. - \frac{H_{\text{left}}[\widehat{\rho}, \rho]}{\Gamma_a} - \frac{H_{\text{right}}[\widehat{\rho}, \rho]}{\Gamma_b} \right] \right\}, \tag{35a}
$$

with the effective Hamiltonian (14c,14d) and

$$h(x,t) = \widehat{\rho}(x,t) + \int_0^x \mathrm{d}y\, \alpha(y). \tag{35b}$$

Compared to (14a) the difference is in the argument of $H_{\mathrm{bulk}}$ and in the lack of a temporal condition on the density field.

For large $L$ and large $T$, the path integral (35a) is dominated by the optimal Action path, which is independent of time. This assertion about stationarity of optimal path independently comes from the additivity conjecture [20, 45]. The optimal Action gives the scgf (34) for large $L$, $\mu(\{\lambda_i\}, \rho_a, \rho_b) \simeq \frac{1}{L}\chi[\alpha(x), \rho_a, \rho_b]$ with

$$\chi[\alpha, \rho_a, \rho_b] = \int_0^1 \mathrm{d}x\, \Big[\rho(x)\big(1-\rho(x)\big)\big(\widehat{\rho}'(x) + \alpha(x)\big) - \rho'(x)\Big]\big(\widehat{\rho}'(x) + \alpha(x)\big)$$
$$+ \frac{\omega\big(\widehat{\rho}(0), \rho_a, \rho(0)\big)}{\Gamma_a} + \frac{\omega\big(\widehat{\rho}(1), \rho_b, \rho(1)\big)}{\Gamma_b}, \tag{36}$$

(the prime (') denotes $\frac{\mathrm{d}}{\mathrm{d}x}$) optimized over time-independent profiles $\widehat{\rho}(x)$ and $\rho(x)$, where we have explicitly written the terms from (35).

The formula (36) reflects an important symmetry when expressed in terms of the $h$-field defined in (35b).

$$\chi[\alpha, \rho_a, \rho_b] \equiv \chi(\lambda, \rho_a, \rho_b) = \int_0^1 \mathrm{d}x\, \Big[\rho(x)\big(1-\rho(x)\big)h'(x) - \rho'(x)\Big]h'(x)$$
$$+ \frac{\omega\big(h(0), \rho_a, \rho(0)\big)}{\Gamma_a} + \frac{\omega\big(h(1)-\lambda, \rho_b, \rho(1)\big)}{\Gamma_b}, \tag{37}$$

optimized over $h(x)$ and $\rho(x)$, where $\lambda = \int_0^1 \mathrm{d}x\, \alpha(x)$ which is also equal to $\sum_{i=0}^L \lambda_i$ from its definition. This means that the scgf depends only on $\lambda$ instead of individual $\lambda_i$'s, which implies that for large $\mathcal{T}$ it does not matter where the current is measured. The same conclusion was drawn from an exact microscopic analysis [81].

Another property we shall shortly use is that the fast coupling limit $\Gamma_{a(b)} \to 0$ of (37) is

$$\chi_{\mathrm{fast}}(\lambda, \rho_a, \rho_b) = \int_0^1 \mathrm{d}x\, \Big[\rho(x)\big(1-\rho(x)\big)h'(x) - \rho'(x)\Big]h'(x), \tag{38}$$

optimized over $h(x)$ and $\rho(x)$ with boundary conditions $\rho(0) = \rho_a$, $\rho(1) = \rho_b$, $h(0) = 0$, and $h(1) = \lambda$. The variational formula (38) is well known [20, 80].

For the marginal boundary coupling, the optimization in (37) is done in parts, separating the bulk and the boundary

$$\chi(\lambda, \rho_a, \rho_b) = \min_{h(0), h(1)} \max_{\rho(0), \rho(1)} \Bigg(\chi_{\mathrm{fast}}\big(h(1)-h(0), \rho(0), \rho(1)\big) + \frac{\omega\big(h(0), \rho_a, \rho(0)\big)}{\Gamma_a}$$
$$+ \frac{\omega\big(h(1)-\lambda, \rho_b, \rho(1)\big)}{\Gamma_b}\Bigg), \tag{39}$$

for real $h$-variables and density in $[0, 1]$, where the relation of the bulk term to (38) is seen by a change of variable $h(x) = \widehat{h}(x) + h(0)$. The same result was derived earlier [48] from an additivity argument.

An exact expression of the scgf for fast coupling $\chi_{\mathrm{fast}}(\lambda, \rho_a, \rho_b) = R_{\mathrm{fast}}\big(\omega(\lambda, \rho_a, \rho_b)\big)$ is known [20, 45, 46] and given in (8), where the dependence on the parameters $\lambda$, $\rho_a$, and $\rho_b$

comes in terms of the function $\omega$. A similar dependence for the marginal boundary coupling is found by simplifying the variational formula (39) to yield $\chi(\lambda, \rho_a, \rho_b) = R\big(\omega(\lambda, \rho_a, \rho_b)\big)$ with $R(\omega)$ in (10b). This non-trivial simplification was discussed in details in [81] and we avoid repeating the analysis here.

The expression of $R(\omega)$ in (10b) can be alternatively written in a parametric formula

$$R(\omega) = \theta^2 + \frac{f_a - 1}{2\Gamma_a} + \frac{f_b - 1}{2\Gamma_b}, \tag{40a}$$

with $\theta$ expressed in terms of $\omega$ by the relation

$$\omega = \sinh^2\theta + \big(\Gamma_a f_b + \Gamma_b f_a\big)\theta\,\sinh(2\theta) + \Big(-1 + 4\,\Gamma_a\,\Gamma_b\,\theta^2 + \frac{1}{2}f_a f_b\Big)\cosh(2\theta), \tag{40b}$$

and $f_{a(b)} = \sqrt{1 + 4\,\theta^2\,\Gamma_{a(b)}^2}$. The formula (40) is similar to an expression reported in [80] for a finite system of arbitrary length $L$.

**Remark:** The subtle reason for minimum instead of maximum over $h$-variable in (39) relates to the contour of $\widehat{\rho}(x,t)$ in (35) being along the imaginary line. This contour is by construction of the path integral formulation in Appendix B where the contour of $\widehat{n}$ in (B.6) is along the imaginary line. For large $L$, the path integral in (35a) is evaluated with the method of steepest descent [90] which gives the scgf in (36) as the Action in (35a) evaluated at its saddle point on the complex $\widehat{\rho}(x,t)$ field and real $\rho(x,t)$ field. The saddle point corresponds to a real value of $\widehat{\rho}(x,t)$ where the path of descent is along the imaginary direction and path of ascent is along the real direction on the complex $\widehat{\rho}$-plane. This means for real values of $\widehat{\rho}(x,t)$ the Action is minimal at the saddle point, which in (39) translates to the minimal over $h$ variable. A simpler example of similar analysis is a derivation of (9) using the MFT.

## 4.1 A solution of the optimal profile

The optimal variables for (37) are solution of

$$\Big[2\rho(x)\big(1-\rho(x)\big)h'(x) - \rho'(x)\Big]' = 0, \tag{41a}$$

$$\big(1 - 2\rho(x)\big)\big(h'(x)\big)^2 + h''(x) = 0, \tag{41b}$$

(the prime ($'$) denotes $\frac{\mathrm{d}}{\mathrm{d}x}$) with the spatial boundary condition

$$\Gamma_a h'(0) = \rho_a\big(e^{h(0)} - 1\big) - (1 - \rho_a)\big(e^{-h(0)} - 1\big), \tag{42a}$$

$$\Gamma_a\Big[2\rho(0)\big(1-\rho(0)\big)h'(0) - \rho'(0)\Big] = \rho_a\big(1-\rho(0)\big)e^{h(0)} - \rho(0)(1-\rho_a)e^{-h(0)}, \tag{42b}$$

at the left boundary $x = 0$ and

$$\Gamma_b h'(1) = (1 - \rho_b)\big(e^{-h(1)+\lambda} - 1\big) - \rho_b\big(e^{h(1)-\lambda} - 1\big), \tag{43a}$$

$$\Gamma_b\Big[2\rho(1)\big(1-\rho(1)\big)h'(1) - \rho'(1)\Big] = \rho(1)(1-\rho_b)e^{-h(1)+\lambda} - \rho_b\big(1-\rho(1)\big)e^{h(1)-\lambda}, \tag{43b}$$

at the right boundary $x = 1$. For the reader, it might be useful to compare the optimal equation and the boundary condition to the stationary case of (16) and (17,18) for the density large deviations problem.

The solution of (41) is obtained following methods similar to [45, 80, 91]. We construct

$$\Big[\rho(x)\big(1-\rho(x)\big(h'(x)\big)^2 - h'(x)\rho'(x)\Big]' = 0, \tag{44}$$

from a linear combination

$$h'(x) \times \Big(\text{l.h.s. of (41a)}\Big) - \rho'(x) \times \Big(\text{l.h.s. of (41b)}\Big) = 0, \tag{45}$$

and following a straightforward algebra. This constriction means, the solution of (41a,44) ensures (41b).

An integration over $x$-variable in (41a,44) gives

$$2\rho(x)\big(1-\rho(x)\big)h'(x) - \rho'(x) = \mathcal{J}, \tag{46}$$

$$\rho(x)\big(1-\rho(x)\big)\big(h'(x)\big)^2 - h'(x)\rho'(x) = \mathcal{H}, \tag{47}$$

where $\mathcal{J}$ and $\mathcal{H}$ are independent of $x$. Comparing with an analogue of (16a) for (35), the constant $\mathcal{J}$ is interpreted as the current in the biased ensemble [81,92] with tilting parameter $\lambda$. Similarly the expression (14c) for $H_{\text{bulk}}$ in (35) shows that the constant $\mathcal{H}$ in (47) is the bulk Hamiltonian for the optimal path.

It is straightforward to verify that a real valued solution of (46-47) is

$$\rho(x) = \frac{1}{2}\left(1 + \frac{\sinh\big\{2\big[\theta_a + (\theta_b - \theta_a)x\big]\big\}}{\sinh(2f)}\right), \tag{48a}$$

with

$$h(x) = c + \log\left[\frac{\cosh\big(f - \theta_a - (\theta_b - \theta_a)x\big)}{\cosh\big(f + \theta_a + (\theta_b - \theta_a)x\big)}\right], \quad \text{for } \lambda(\rho_a - \rho_b) > 0, \tag{48b}$$

and

$$h(x) = c + \log\left[\frac{\sinh\big(f + \theta_a + (\theta_b - \theta_a)x\big)}{\sinh\big(f - \theta_a - (\theta_b - \theta_a)x\big)}\right], \quad \text{for } \lambda(\rho_a - \rho_b) < 0, \tag{48c}$$

with constants $c$, $\theta_a$, $\theta_b$, and $f$ such that

$$(\theta_a - \theta_b)^2 = \mathcal{H}, \quad \text{and} \quad (\theta_a - \theta_b)\coth(2f)\,\text{sgn}\big(\lambda(\rho_a - \rho_b)\big) = \mathcal{J}. \tag{48d}$$

Therefore, (48) is the solution of the Euler-Lagrange equation (41) with the parameters $c$, $\theta_a$, $\theta_b$, and $f$ determined from the boundary condition (42,43). The above solution for $h(x)$ yields real values for the parameters. The optimal profile for different parameter values is shown in Fig. 4 for the fast coupling regime and in Fig. 5 for the marginal boundary coupling.

**Remark**: The scgf of current in (37) can be expressed in terms of the parameters in (48).

$$\chi(\lambda, \rho_a, \rho_b) = (\theta_a - \theta_b)^2 + \frac{B(\theta_a, c, \rho_a)}{\Gamma_a} + \frac{B(\theta_b, c - \lambda, \rho_b)}{\Gamma_b}, \tag{49a}$$

with

$$B(\theta, c, \rho) = \frac{\cosh(\theta + f) - e^c \cosh(\theta - f)}{\sinh(2f)}\big(\rho \sinh(\theta - f) + e^{-c}(1 - \rho)\sinh(\theta + f)\big), \tag{49b}$$

for $\lambda(\rho_a - \rho_b) > 0$. For the other parameter regime $\lambda(\rho_a - \rho_b) < 0$, a similar solution can be constructed.

**Remark**: In the fast coupling limit $\Gamma_{a(b)} \to 0$, the boundary condition (42,43) reduces to a Dirichlet condition $\rho(0) = \rho_a$, $\rho(1) = \rho_b$, $h(0) = 0$, and $h(1) = \lambda$. For the solution in (48), the boundary condition on $h(0)$ and $h(1)$ give $\cosh(\theta_a + f) = e^c \cosh(\theta_a - f)$ and $\cosh(\theta_b + f) = e^{c - \lambda}\cosh(\theta_b - f)$, respectively, leading to vanishing of the boundary terms in (49a) and the scgf $\chi(\lambda, \rho_a, \rho_b) = (\theta_a - \theta_b)^2$ as reported in [45].

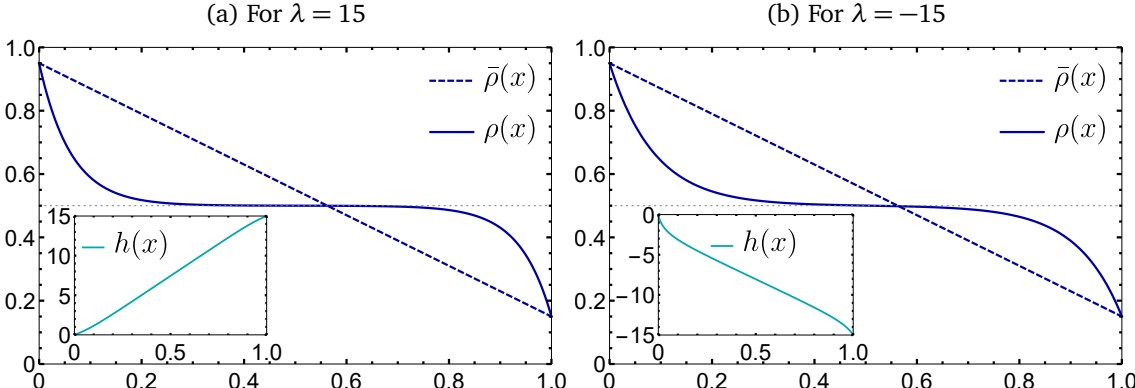

Figure 4: **Fast coupling regime** – The optimal quasi-stationary density profile $\rho(x)$ in (48) for $\rho_a = 0.95$, $\rho_b = 0.15$ in the fast coupling limit ($\Gamma_{a(b)} \to 0$) with the values of $\lambda$ indicated. Even though the two density profiles may appear similar on the shown plot range, they are different. The stationary average density profile $\bar{\rho}(x)$ (corresponds to $\lambda = 0$) is shown in dashed line for a reference. The insets show corresponding profile for the optimal conjugate field $h(x)$.

# 5 Conclusion

We revisited a well-known problem about the SSEP on a one-dimensional finite lattice coupled with two boundary reservoirs of unequal densities. In seminal works by Derrida and collaborators [43, 45, 46] the fluctuations of density and current in the non-equilibrium stationary state of the model were characterized in terms of corresponding large deviation functions. These results were subsequently reproduced using the powerful hydrodynamic approach of the MFT by Bertini, De Sole, Gabrielli, Jona-Lasinio, and Landim [20, 39]. Recently, these large deviation results were generalized [48] to incorporate situations where coupling to the reservoirs is slow, specifically, when the boundary rates are of the order of inverse of the system size. In this article, we recovered these extended results using the hydrodynamic approach of the MFT.

The slow coupling makes fluctuations at the boundary comparable to bulk fluctuations, which modifies the fluctuating hydrodynamics (see (12a)) for the SSEP and introduces additional terms (see (14)) in the corresponding MSRJD-Action [85–88]. In the MFT formulation for large deviations, these additional terms significantly alter the boundary conditions of the hydrodynamic fields, posing challenges in solving the corresponding Euler-Lagrange equations. For the large deviations of density, we tackled (see Section 3) this challenge by employing a simple *local* transformation (19) and successfully recovered the earlier result [48]. Similarly, for the large deviations of current [48], we utilized the quasi-stationarity of optimal path (see Section 4) to recover the results. Our explicit solutions for the optimal path in both the density-problem and current-problems provide insights into how rare fluctuations develop in the non-equilibrium stationary state.

The local transformation (19) and the method outlined in Section 3.1 for solving the variational problem of density fluctuations can be extended to address similar variational problems in related models. This includes the weakly asymmetric exclusion process [63, ], symmetric simple partial exclusion process [94, 95], the zero range process [96, 97], the symmetric simple inclusion process [98, 99], and the Kipnis-Marchioro-Presutti model [100, 101]. For an even broader class of models, the variational problem can be tackled perturbatively around equilibrium. These extensions will be detailed in a forthcoming publication [102]. It would be interesting to see whether similar approach could be used in higher dimensions [71] and for

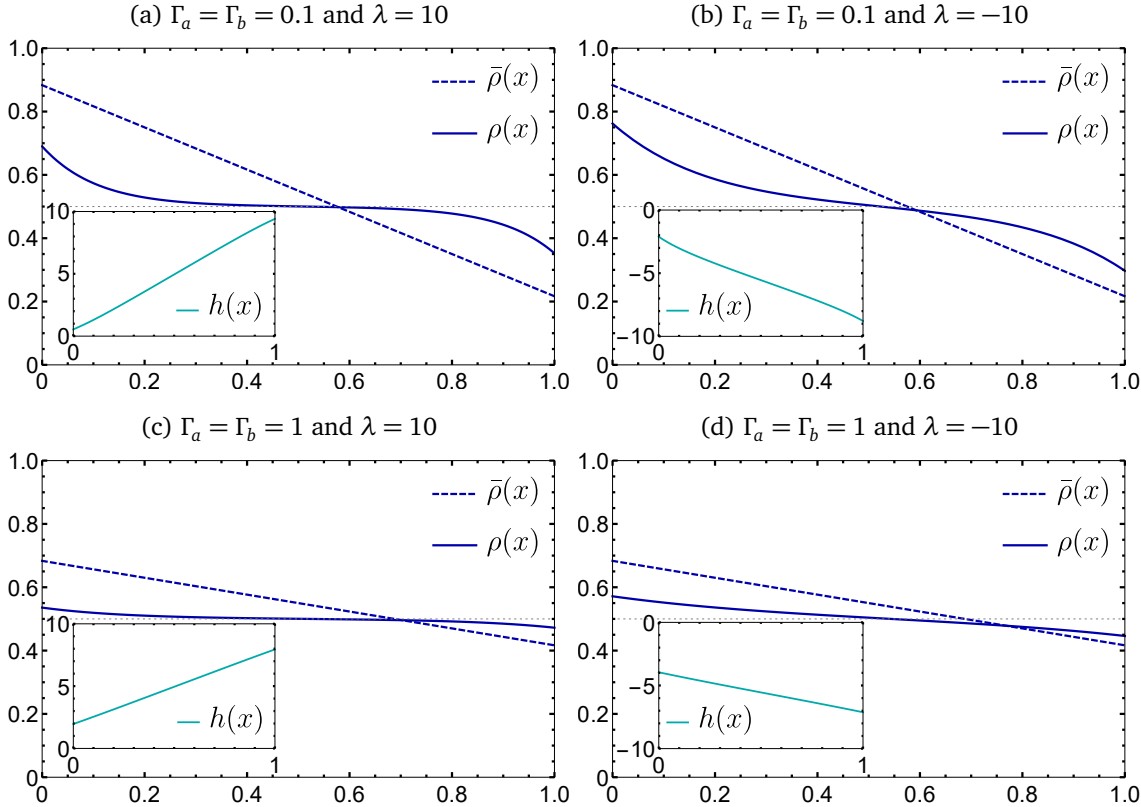

Figure 5: **Marginal boundary coupling** – The optimal quasi-stationary density profile $\rho(x)$ in (48) with the associated boundary condition (42-43) for $\rho_a = 0.95$, $\rho_b = 0.15$, and rest of the parameters values indicated above the figures. The stationary average density profile $\bar{\rho}(x)$ is shown in dashed line for a reference. The insets show corresponding profile for the optimal conjugate field $h(x)$.

systems with multiple coupled hydrodynamic fields, such as the active lattice gas [103–105]. Another intriguing and challenging problem is the large deviation of density conditioned on current [81], where similar solution methods may prove useful. In fact, the effect of slow bonds on current fluctuations on infinite and semi-infinite geometry is an interesting topic, where techniques of MFT offer exact solutions [106].

# Acknowledgements

TS acknowledges insightful discussions with Bernard Derrida about this work. Many crucial ideas, especially the importance of considering the slow coupling, the derivation of the hydrodynamic Action, solution of the variational problem, and the Hamilton-Jacobi equation for marginal boundary coupling, have originated from those discussions. TS particularly acknowledges the lecture notes from Bernard Derrida delivered at the College-de-France in 2017, which immensely helped this work. The non-triviality of slow coupling was first brought to our attention by Ori Hirschberg several years ago in late 2012.

**Funding information** We acknowledge financial support of the Department of Atomic Energy, Government of India, under Project Identification No. RTI 4002.

# A  Hydrodynamics for the SSEP with slow boundaries: A microscopic derivation

For the dynamics of SSEP with marginal boundary coupling defined in Fig. 2 the average occupation $\langle n_i(\tau) \rangle$ of the $i$-th site at a microscopic time $\tau$ follows the rate equation [1, 84]

$$\frac{\mathrm{d} \langle n_i(\tau) \rangle}{\mathrm{d}\tau} = -\mathcal{A}_{ij} \langle n_j(\tau) \rangle + \mathcal{B}_i, \qquad \text{with } 1 \le i, j \le L, \tag{A.1a}$$

(following the Einstein summation notation) where the operators $\mathcal{A}$ and $\mathcal{B}$ are respectively given by

$$\mathcal{A} = \begin{pmatrix} 1+\gamma_a & -1 & 0 & \cdots & \cdots & \cdots & \cdots & 0 \\ -1 & 2 & -1 & 0 & \cdots & \cdots & \cdots & 0 \\ \vdots & \vdots & \vdots & \vdots & \vdots & \vdots & \vdots & \vdots \\ 0 & \cdots & \cdots & \cdots & 0 & -1 & 2 & -1 \\ 0 & \cdots & \cdots & \cdots & \cdots & 0 & -1 & 1+\gamma_b \end{pmatrix}, \text{ and } \mathcal{B} = \begin{pmatrix} \gamma_a \rho_a \\ 0 \\ \vdots \\ 0 \\ \gamma_b \rho_b \end{pmatrix}, \tag{A.1b}$$

with $\gamma_a = 1/(\Gamma_a L)$ and $\gamma_b = 1/(\Gamma_b L)$.

At large times, the average occupation asymptotically reaches a stationary value $\bar{n}_i$ which is a solution of

$$\mathcal{A}_{ij} \bar{n}_j = \mathcal{B}_i. \tag{A.2}$$

The difference $\langle \widetilde{n}_i(\tau) \rangle = \langle n_i(\tau) \rangle - \bar{n}_i$ follows a homogeneous equation

$$\frac{\mathrm{d}\langle \widetilde{n}_i(\tau) \rangle}{\mathrm{d}\tau} + \mathcal{A}_{ij} \langle \widetilde{n}_j(\tau) \rangle = 0. \tag{A.3}$$

The solution of (A.2) is a linear function [1, 84]

$$\bar{n}_i = \rho_a \left( 1 - \frac{i - 1 + \frac{1}{\gamma_a}}{L - 1 + \frac{1}{\gamma_a} + \frac{1}{\gamma_b}} \right) + \rho_b \frac{i - 1 + \frac{1}{\gamma_a}}{L - 1 + \frac{1}{\gamma_a} + \frac{1}{\gamma_b}}. \tag{A.4}$$

The solution of (A.3) can be expressed in terms of the eigenvalues and the corresponding eigenvectors of the operator $\mathcal{A}$. For the $L \times L$ symmetric matrix $\mathcal{A}$ there are $L$ number of distinct real eigenvalues $\mathcal{E}_k = 4 \sin^2 \frac{\Lambda_k}{2}$ with the corresponding eigenvector $\Psi^{(k)}$ whose components

$$\Psi_i^{(k)} = \sin(\Lambda_k i - \mu_k) \sqrt{\frac{2}{L - (-1)^k \cos(\mu_k - \nu_k) \frac{\sin \Lambda_k L}{\sin \Lambda_k}}}, \tag{A.5a}$$

for $k = 1, 2, \cdots, L$ where the parameters $\Lambda_k$ are solutions of the transcendental equation

$$(L + 1) \Lambda_k = k\pi + \mu_k + \nu_k, \tag{A.5b}$$

with

$$\tan \mu_k = \frac{(\gamma_a - 1) \sin \Lambda_k}{1 + (\gamma_a - 1) \cos \Lambda_k} \qquad \text{and} \qquad \tan \nu_k = \frac{(\gamma_b - 1) \sin \Lambda_k}{1 + (\gamma_b - 1) \cos \Lambda_k}. \tag{A.5c}$$

A formal spectral solution of (A.3) is

$$\langle \widetilde{n}_i(\tau) \rangle = \sum_{k=1}^{L} e^{-4 \sin^2 \frac{\Lambda_k}{2} \tau} \Psi_i^{(k)} \sum_{j=1}^{L} \left[ \Psi_j^{(k)} \right]^\star \langle \widetilde{n}_j(0) \rangle. \tag{A.6}$$

For large $L$, the solution (A.4) has the asymptotic $\bar{n}_i \simeq \bar{\rho}\left(\frac{i}{L}\right)$ given in (32) which is a fixed point of the diffusion equation with Robin boundary condition (11b). Similarly, the solution (A.6) has the asymptotic $\langle \tilde{n}_i(\tau) \rangle \simeq \delta\bar{\rho}\left(\frac{i}{L}, \frac{\tau}{L^2}\right)$ with

$$\delta\bar{\rho}(x,t) = 2\sum_{k=1}^{\infty}\left[\frac{e^{-\lambda_k^2 t}\sin(\lambda_k x + \mu_k)}{1-(-1)^k\cos(\mu_k - \nu_k)\frac{\sin\lambda_k}{\lambda_k}}\int_0^1 dy\ \sin(\lambda_k y + \mu_k)\,\delta\bar{\rho}(y,0)\right], \quad \text{(A.7a)}$$

where $\lambda_k$ are solutions of the transcendental equation

$$\tan\lambda_k = \frac{(\Gamma_a + \Gamma_b)\lambda_k}{\Gamma_a\Gamma_b\lambda_k^2 - 1}, \quad \text{(A.7b)}$$

with

$$\mu_k = \tan^{-1}\Gamma_a\lambda_k \quad \text{and} \quad \nu_k = \tan^{-1}\Gamma_b\lambda_k, \quad \text{(A.7c)}$$

Similar to $\bar{\rho}(x)$ it is straightforward to verify that $\delta\bar{\rho}(x,t)$ in (A.7) satisfies the diffusion equation with the Robin boundary condition

$$\delta\bar{\rho}(0,t) - \Gamma_a\,\partial_x\delta\bar{\rho}(0,t) = 0, \quad \text{and} \quad \delta\bar{\rho}(1,t) + \Gamma_b\,\partial_x\delta\bar{\rho}(1,t) = 0. \quad \text{(A.8)}$$

This means, the hydrodynamic limit $\langle n_i(\tau) \rangle \simeq \bar{\rho}\left(\frac{i}{L}, \frac{\tau}{L^2}\right)$ with $\bar{\rho}(x,t) = \bar{\rho}(x) + \delta\bar{\rho}(x,t)$ is described by the diffusion equation with the Robin boundary condition in (11).

**Remark**: The fast and the slow coupling limits can be similarly analyzed using $\gamma_{a(b)} = 1/L^\theta$ in (A.1b). For the fast coupling ($\theta < 1$) we get

$$\bar{\rho}(x,t) = \rho_a(1-x) + \rho_b x + 2\sum_{k=1}^{\infty}\int_0^1 dy\left(e^{-\lambda_k^2 t}\sin(\lambda_k x)\sin(\lambda_k y)\,\delta\bar{\rho}(y,0)\right), \quad \text{(A.9a)}$$

with $\lambda_k = k\pi$, whereas for the slow coupling ($\theta > 1$)

$$\bar{\rho}(x,t) = \frac{\rho_a + \rho_b}{2} + 2\sum_{k=1}^{\infty}\int_0^1 dy\left(e^{-\lambda_k^2 t}\cos(\lambda_k x)\cos(\lambda_k y)\,\delta\bar{\rho}(y,0)\right), \quad \text{(A.9b)}$$

with $\lambda_k = (k+1)\pi$. The uniform stationary density in (A.9b) for the slow coupling asserts that the stationary state is effectively in equilibrium, being in agreement with (2).

# B  A derivation of the complete MFT-Action for the SSEP with slow boundaries

The path integral description (14a) for the fluctuating hydrodynamics of SSEP is well-known [37, 39–41, 47] in the fast coupling limit. Even for the slow coupling, earlier discussions can be found in [41, 80, 83, 107]. Here we present a derivation that is not as rigorous as one would desire, but we believe it is simpler to follow for a general Physics reader. To our knowledge, a derivation along this line appeared earlier in [107]. A different approach for the fluctuating hydrodynamics of SSEP is discussed in [84].

For the SSEP defined in Fig. 2, $n_i(\tau)$ denotes the occupancy of the $i$-th site at time $\tau$ while $Y_i(\tau)$ denotes the rightward flux of particles across the bond connecting the $i$-th and the $i+1$-th sites between times $\tau$ and $\tau + d\tau$. For infinitesimally small $d\tau$ the observable $Y_i(\tau)$ takes

values ±1 and 0. Their probability across the bonds in the bulk of the lattice ($1 \leq i \leq L - 1$) is as follows.

$$
Y_i(\tau) = \begin{cases} 1 & \text{with prob. } n_i(\tau)\big(1 - n_{i+1}(\tau)\big)\,\mathrm{d}\tau\,, & \text{(B.1a)} \\ -1 & \text{with prob. } n_{i+1}(\tau)\big(1 - n_i(\tau)\big)\,\mathrm{d}\tau\,, & \text{(B.1b)} \\ 0 & \text{with prob. } 1 - \Big[n_i(\tau)\big(1 - n_{i+1}(\tau)\big) + n_{i+1}(\tau)\big(1 - n_i(\tau)\big)\Big]\mathrm{d}\tau\,. & \text{(B.1c)} \end{cases}
$$

Across the bond linking the first lattice site ($i = 1$) with the left reservoir

$$
Y_0(\tau) = \begin{cases} 1 & \text{with prob. } \gamma_a\,\rho_a\big(1 - n_1(\tau)\big)\,\mathrm{d}\tau\,, & \text{(B.2a)} \\ -1 & \text{with prob. } \gamma_a\,n_1(\tau)(1 - \rho_a)\,\mathrm{d}\tau\,, & \text{(B.2b)} \\ 0 & \text{with prob. } 1 - \gamma_a\Big[\rho_a\big(1 - n_1(\tau)\big) + n_1(\tau)(1 - \rho_a)\Big]\mathrm{d}\tau\,, & \text{(B.2c)} \end{cases}
$$

while across the bond linking the last lattice site ($i = L$) with the right reservoir

$$
Y_L(\tau) = \begin{cases} 1 & \text{with prob. } \gamma_b\,n_L(\tau)(1 - \rho_b)\,\mathrm{d}\tau\,, & \text{(B.3a)} \\ -1 & \text{with prob. } \gamma_b\,\rho_b\big(1 - n_L(\tau)\big)\,\mathrm{d}\tau\,, & \text{(B.3b)} \\ 0 & \text{with prob. } 1 - \gamma_b\Big[n_L(\tau)(1 - \rho_b) + \rho_b\big(1 - n_L(\tau)\big)\Big]\mathrm{d}\tau\,, & \text{(B.3c)} \end{cases}
$$

where we denote $\gamma_a = 1/(\Gamma_a L)$ and $\gamma_b = 1/(\Gamma_b L)$.

The dynamics inside the bulk locally preserves the number of particles. This conservation is stated in a relation

$$
n_i(\tau + \mathrm{d}\tau) - n_i(\tau) = Y_{i-1}(\tau) - Y_i(\tau)\,, \tag{B.4}
$$

for all $1 \leq i \leq L$.

The configuration of the system at any time $\tau$ is specified by $\mathbf{n}(\tau) \equiv \{n_1(\tau), \cdots, n_L(\tau)\}$. Its evolution is governed by the sequence of jump events $\{Y_0(\tau), \cdots, Y_L(\tau)\}$. We denote the initial configuration at $\tau = 0$ by $\mathbf{n}_{\mathrm{ini}} \equiv \mathbf{n}(0)$ and the final configuration at $\tau = \mathcal{T}$ by $\mathbf{n}_{\mathrm{fin}} \equiv \mathbf{n}(\mathcal{T})$. The transition probability $\mathcal{P} \equiv \Pr[\mathbf{n}_{\mathrm{ini}} \to \mathbf{n}_{\mathrm{fin}}]$ is the sum of probability of all evolutions between $\mathbf{n}_{\mathrm{ini}}$ and $\mathbf{n}_{\mathrm{fin}}$ which we formally write

$$
\mathcal{P} = \int_{\mathbf{n}_{\mathrm{ini}}}^{\mathbf{n}_{\mathrm{fin}}} [\mathcal{D}n] \Big\langle \prod_{k=0}^{M-1} \prod_{i=1}^{L} \delta_{n_i(k\,\mathrm{d}\tau + \mathrm{d}\tau) - n_i(k\,\mathrm{d}\tau),\, Y_{i-1}(k\,\mathrm{d}\tau) - Y_i(k\,\mathrm{d}\tau)} \Big\rangle_Y\,, \tag{B.5a}
$$

where $\delta_{a,b}$ is the Kronecker delta function, $M\mathrm{d}\tau = \mathcal{T}$ with and infinitesimal $\mathrm{d}\tau$, and the path integral measure

$$
\int [\mathcal{D}n] \equiv \prod_{k=1}^{M-1} \prod_{i=1}^{L} \sum_{n_i(k\,\mathrm{d}\tau)=0}^{1}\,. \tag{B.5b}
$$

The angular bracket $\langle\rangle_Y$ denotes average over all jump events $\{Y_i(\tau)\}$ in the duration $\mathcal{T}$.

Using an integral representation $\delta_{a,b} = (2\pi\mathrm{i})^{-1} \int_{-\mathrm{i}\pi}^{\mathrm{i}\pi} \mathrm{d}z\, e^{-z(a-b)}$ for integers $a$ and $b$, we write

$$
\mathcal{P} = \int_{\mathbf{n}_{\mathrm{ini}}}^{\mathbf{n}_{\mathrm{fin}}} [\mathcal{D}n][\mathcal{D}\widehat{n}]\, e^{-\sum_{k=0}^{M-1} \sum_{i=1}^{L} \widehat{n}_i(k\,\mathrm{d}\tau)\big(n_i(k\,\mathrm{d}\tau + \mathrm{d}\tau) - n_i(k\,\mathrm{d}\tau)\big)}
$$

$$
\Big\langle e^{\sum_{k=0}^{M-1} \sum_{i=1}^{L} \widehat{n}_i(k\,\mathrm{d}\tau)\big(Y_{i-1}(k\,\mathrm{d}\tau) - Y_i(k\,\mathrm{d}\tau)\big)} \Big\rangle_Y\,, \tag{B.6}
$$

with the path integral measure on $\widehat{n}$ defined accordingly. In this new form the average over $Y$ is easy to compute using the probability (B.1, B.2, B.3). For this we separate the boundary terms from the bulk terms in the exponent by writing for each time-index $k$,

$$
\sum_{i=1}^{L} \widehat{n}_i\big(Y_{i-1} - Y_i\big) = \sum_{i=1}^{L-1} \big(\widehat{n}_{i+1} - \widehat{n}_i\big) Y_i + \widehat{n}_1 Y_0 - \widehat{n}_L Y_L\,. \tag{B.7}
$$

For the bulk terms ($1 \leq i \leq L-1$), using (B.1), the average for each time-index $k$ yields

$$\left\langle e^{(\widehat{n}_{i+1}-\widehat{n}_i)Y_i} \right\rangle_{Y_i} \simeq \exp\left\{ d\tau \left[ \left( e^{\widehat{n}_{i+1}-\widehat{n}_i} - 1 \right) n_i \left( 1 - n_{i+1} \right) + \left( e^{-\widehat{n}_{i+1}+\widehat{n}_i} - 1 \right) n_{i+1} \left( 1 - n_i \right) \right] \right\}, \quad \text{(B.8)}$$

where the $\simeq$ denotes the leading term for vanishing $d\tau$ limit. Similarly, using (B.2) and (B.3) the average of the boundary terms for each time-index $k$,

$$\left\langle e^{\widehat{n}_1 Y_0} \right\rangle_{Y_0} \simeq \exp\left\{ \gamma_a \, d\tau \left[ \left( e^{\widehat{n}_1} - 1 \right) \rho_a \left( 1 - n_1 \right) + \left( e^{-\widehat{n}_1} - 1 \right) n_1 \left( 1 - \rho_a \right) \right] \right\}, \quad \text{(B.9)}$$

$$\left\langle e^{-\widehat{n}_L Y_L} \right\rangle_{Y_L} \simeq \exp\left\{ \gamma_b \, d\tau \left[ \left( e^{\widehat{n}_L} - 1 \right) \rho_b \left( 1 - n_L \right) + \left( e^{-\widehat{n}_L} - 1 \right) n_L \left( 1 - \rho_b \right) \right] \right\}. \quad \text{(B.10)}$$

Using these results for averages (B.8-B.10) in the path integral (B.6), we write in the vanishing $d\tau$ limit,

$$\mathcal{P} = \int_{\mathbf{n}_{\text{ini}}}^{\mathbf{n}_{\text{fin}}} [\mathcal{D}n][\mathcal{D}\widehat{n}] \, e^{\mathcal{K}+\mathcal{H}_{\text{bulk}}+\mathcal{H}_{\text{left}}+\mathcal{H}_{\text{right}}}, \quad \text{(B.11a)}$$

where

$$\mathcal{K} = \sum_{i=1}^{L}\left[ \widehat{n}_i(0)\, n_i(0) - \widehat{n}_i(\mathcal{T})\, n_i(\mathcal{T}) + \int_0^{\mathcal{T}} d\tau \left( \frac{d\widehat{n}_i(\tau)}{d\tau} n_i(\tau) \right) \right], \quad \text{(B.11b)}$$

$$\mathcal{H}_{\text{bulk}} = \sum_{i=1}^{L-1} \int_0^{\mathcal{T}} d\tau \, \omega\big( \widehat{n}_{i+1}(\tau) - \widehat{n}_i(\tau), \, n_i(\tau), \, n_{i+1}(\tau) \big), \quad \text{(B.11c)}$$

$$\mathcal{H}_{\text{left}} = \gamma_a \int_0^{\mathcal{T}} d\tau \, \omega\big( \widehat{n}_1(\tau), \, \rho_a, n_1(\tau) \big), \quad \text{(B.11d)}$$

$$\mathcal{H}_{\text{right}} = \gamma_b \int_0^{\mathcal{T}} d\tau \, \omega\big( \widehat{n}_L(\tau), \, \rho_b, n_L(\tau) \big), \quad \text{(B.11e)}$$

with $\omega(\lambda, x, y)$ defined in (7). In writing (B.11b) we have rearranged the terms in the first exponential in (B.6) such that

$$\mathcal{K} = \sum_{i=1}^{L}\left[ \widehat{n}_i(0)\, n_i(0) - \widehat{n}_i(\mathcal{T}-d\tau)\, n_i(\mathcal{T}) + \sum_{k=1}^{M-1} \big( \widehat{n}_i(k\,d\tau) - \widehat{n}_i(k\,d\tau - d\tau) \big) n_i(k\,d\tau) \right], \quad \text{(B.11f)}$$

and assumed that $\widehat{n}_i(\tau)$ is continuous with finite time-derivative.

**Remark**: For the path integral (B.11a) each of the terms in the Action (B.11b-B.11e) are to be interpreted with discretized time $d\tau$, *e.g.*, (B.11f) for (B.11b), and rest of the time integrals with Itô convention.

## B.1 Hydrodynamic limit

For a diffusive system like the SSEP, at large times of order $L^2$, fluctuations in regions of length of order $L$ effectively reach a local equilibrium. For the SSEP this means, the local distribution of occupation variables $n_i(\tau)$ can be approximated by the equilibrium measure corresponding to a smoothly varying average density

$$\rho_i(\tau) \simeq \rho\left( \frac{i}{L}, \frac{\tau}{L^2} \right), \quad \text{(B.12)}$$

for large $L$. The hydrodynamic description is about this smoothly varying density $\rho(x, t)$.

With the assumption of local equilibrium and a smoothly varying conjugate field

$$\widehat{n}_i(\tau) \simeq \widehat{\rho}\left(\frac{i}{L}, \frac{\tau}{L^2}\right), \tag{B.13}$$

the leading term of the transition probability (B.11a) for large $L$ can be written in terms of $\rho_i(\tau)$ by averaging with the local equilibrium measure which for the SSEP is a Bernoulli distribution

$$n_i(\tau) = \begin{cases} 1 & \text{with probability } \rho_i(\tau), & \text{(B.14a)} \\ 0 & \text{with probability } 1 - \rho_i(\tau). & \text{(B.14b)} \end{cases}$$

We are interested in the transition probability between an initial configuration $n_i(0)$ and a final configuration $n_i(\mathcal{T})$ which are typical of the smooth density profiles $\rho_i(0)$ and $\rho_i(\mathcal{T})$, respectively, such that

$$\sum_i n_i(0)\widehat{n}_i(0) \simeq \sum_i \rho_i(0)\widehat{n}_i(0) \quad \text{and} \quad \sum_i n_i(\mathcal{T})\widehat{n}_i(\mathcal{T}) \simeq \sum_i \rho_i(\mathcal{T})\widehat{n}_i(\mathcal{T}), \tag{B.15}$$

for large $L$ and smooth functions (B.12,B.13). The transition probability for large $L$,

$$\mathcal{P} \simeq \int_{\boldsymbol{\rho}_{\text{ini}}}^{\boldsymbol{\rho}_{\text{fin}}} [\mathcal{D}\rho][\mathcal{D}\widehat{n}] \left\langle e^{\mathcal{K} + \mathcal{H}_{\text{bulk}} + \mathcal{H}_{\text{left}} + \mathcal{H}_{\text{right}}} \right\rangle_{\mathbf{n}}, \tag{B.16}$$

where $\boldsymbol{\rho}_{\text{ini}} \equiv \{\rho_i(0)\}$ and $\boldsymbol{\rho}_{\text{fin}} \equiv \{\rho_i(\mathcal{T})\}$.

The task of averaging in (B.16) simplifies using that for large $L$ the probability (B.14) at different times can assumed to be independent, which leads to

$$\mathcal{P} \simeq \int_{\boldsymbol{\rho}_{\text{ini}}}^{\boldsymbol{\rho}_{\text{fin}}} [\mathcal{D}\rho][\mathcal{D}\widehat{n}] e^{\sum_{i=1}^{L}\left(\widehat{n}_i(0)n_i(0) - \widehat{n}_i(\mathcal{T})n_i(\mathcal{T})\right)} \left(\prod_\tau \left\langle e^{d\tau\, s(\tau)} \right\rangle_{\mathbf{n}(\tau)}\right), \tag{B.17a}$$

with

$$s(\tau) = \sum_{i=1}^{L}\left(\frac{d\widehat{n}_i(\tau)}{d\tau} n_i(\tau)\right) + \sum_{i=1}^{L-1}\omega\left(\widehat{n}_{i+1}(\tau) - \widehat{n}_i(\tau), n_i(\tau), n_{i+1}(\tau)\right)$$
$$+ \gamma_a\, \omega\left(\widehat{n}_1(\tau), \rho_a, n_1(\tau)\right) + \gamma_b\, \omega\left(\widehat{n}_L(\tau), \rho_b, n_L(\tau)\right). \tag{B.17b}$$

In the vanishing $d\tau$ limit,

$$\left\langle e^{d\tau\, s(\tau)} \right\rangle \simeq 1 + d\tau \left\langle s(\tau) \right\rangle \simeq e^{d\tau\, \langle s(\tau) \rangle}. \tag{B.18}$$

Average of the terms in (B.17b) is simple to get using that the probability (B.14) is independent for different sites. The average $\left\langle s(\tau) \right\rangle_{\mathbf{n}(\tau)}$ has a similar expression as in (B.17b) with $n_i(\tau)$ replaced by $\rho_i(\tau)$ for all sites $i$. This gives the transition probability (B.17a)

$$\mathcal{P} \simeq \int_{\boldsymbol{\rho}_{\text{ini}}}^{\boldsymbol{\rho}_{\text{fin}}} [\mathcal{D}\widehat{n}][\mathcal{D}\rho] e^{\sum_{i=1}^{L}\left[\widehat{n}_i(0)\left(n_i(0) - \rho_i(0)\right) - \widehat{n}_i(\mathcal{T})\left(n_i(\mathcal{T}) - \rho_i(\mathcal{T})\right)\right]} e^{S[\widehat{n},\rho]}, \tag{B.19a}$$

with

$$S[\widehat{n}, \rho] = \int_0^{\mathcal{T}} d\tau \left\{ -\sum_{i=1}^{L}\left(\widehat{n}_i(\tau)\frac{d\rho_i(\tau)}{d\tau}\right) + \sum_{i=1}^{L-1}\omega\left(\widehat{n}_{i+1}(\tau) - \widehat{n}_i(\tau), \rho_i(\tau), \rho_{i+1}(\tau)\right) \right.$$
$$\left. + \gamma_a\, \omega\left(\widehat{n}_1(\tau), \rho_a, \rho_1(\tau)\right) + \gamma_b\, \omega\left(\widehat{n}_L(\tau), \rho_b, \rho_L(\tau)\right) \right\}, \tag{B.19b}$$

where we used an integration by parts in the time variable to get the first term in (B.19b) such that the boundary terms involving $\rho_i(0)$ and $\rho_i(\mathcal{T})$ goes inside the first exponential in (B.19a).

For large $L$, the first exponential in (B.19a) contributes value one due to the condition (B.15). For the second exponential, we write $S$ in terms of the hydrodynamic fields $\rho(x,t)$ and $\widehat{\rho}(x,t)$ in (B.12) and (B.13), and then using a gradient expansion for large $L$ we obtain the hydrodynamic transition probability (14).

## C Fluctuating hydrodynamics for the SSEP with slow boundaries: A derivation of (12-13)

The Action in (14a) is the Martin-Siggia-Rose-Janssen-De Dominicis Action [85–88] of the stochastic differential equation (12) with $\widehat{\rho}$ being the response field. One simple way to see this is by writing the transition probability between density profiles $\rho(x,t_i)$ and $\rho(x,t_f)$ governed by the dynamics (12) as

$$
\mathcal{P} = \int_{\rho(x,t_i)}^{\rho(x,t_f)} [\mathcal{D}\rho] \left\langle \prod_{t,x} \delta\big(\partial_x^2\rho(x,t) + \partial_x\eta(x,t) - \partial_t\rho(x,t)\big) \right.
$$
$$
\prod_t \delta\big(\partial_x\rho(0,t) + \eta(0,t) + \xi_{\text{left}}(t)\big)
$$
$$
\left. \prod_t \delta\big(-\partial_x\rho(1,t) - \eta(1,t) - \xi_{\text{right}}(t)\big) \right\rangle_{\{\eta,\xi_{\text{left}},\xi_{\text{right}}\}}, \qquad \text{(C.1)}
$$

where the angular bracket denotes averages over realization of the noises $\eta(x,t)$, $\xi_{\text{left}}(t)$ and $\xi_{\text{right}}(t)$.

Using an integral representation of the Dirac delta function $\delta(y) = (2\pi i)^{-1} \int_{-\pi i}^{\pi i} dz\, e^{yz}$ with the contour along the imaginary line, we introduce the response field $\widehat{\rho}(x,t)$ at every space-time point in (C.1). Using an integration by parts in the space variable, and demanding that $\widehat{\rho}(x,t)$ is a smooth function, we get

$$
\mathcal{P} = \int_{\rho(x,t_i)}^{\rho(x,t_f)} [\mathcal{D}\widehat{\rho}][\mathcal{D}\rho]\, e^{-L\int_{t_i}^{t_f} dt \int_0^1 dx \left(\widehat{\rho}(x,t)\partial_t\rho(x,t) + \partial_x\widehat{\rho}(x,t)\partial_x\rho(x,t)\right)}
$$
$$
\left\langle e^{L\int_{t_i}^{t_f} dt \int_0^1 dx\, \eta(x,t)\partial_x\widehat{\rho}(x,t)} \right\rangle_\eta \left\langle e^{L\int_{t_i}^{t_f} dt\, \widehat{\rho}(0,t)\xi_{\text{left}}(t)} \right\rangle_{\xi_{\text{left}}} \left\langle e^{-L\int_{t_i}^{t_f} dt\, \widehat{\rho}(1,t)\xi_{\text{right}}(t)} \right\rangle_{\xi_{\text{right}}}. \quad \text{(C.2)}
$$

Completing the average over the Gaussian noise $\eta(x,t)$ with covariance (13a) and the averages over $\xi_{\text{left}}$ and $\xi_{\text{right}}$ using (13b-13c) it is straightforward to arrive at (14) where $t_i \to -\infty$ with $\rho(x,t_i) = \bar{\rho}(x)$ and $t_f = 0$ with $\rho(x,t_f) = r(x)$.

A more careful analysis done by discretizing (12a) with the Itô convention also confirms (14).

## D An explicit verification that the solution (23-24) satisfies the Euler-Lagrange equation (20a)

Starting with the identity (24) that relates $\rho(x,t)$ to $F(x,t)$, we separately take a first-order time derivative and a second-order space derivative and then add the two resulting equations.

This gives

$$
\partial_t \rho(x,t) + \partial_x^2 \rho(x,t) = \partial_t F(x,t) + \partial_x^2 F(x,t) + \partial_t \left[ \frac{F(x,t)\left(1-F(x,t)\right)\partial_x^2 F(x,t)}{\left(\partial_x F(x,t)\right)^2} \right]
$$
$$
+ \partial_x^2 \left[ \frac{F(x,t)\left(1-F(x,t)\right)\partial_x^2 F(x,t)}{\left(\partial_x F(x,t)\right)^2} \right]. \tag{D.1}
$$

The RHS simplifies by expressing the temporal derivative of $F$ in terms of spatial derivatives of the function using (23) leading to

$$
\partial_t \rho(x,t) + \partial_x^2 \rho(x,t) = \frac{\left(1-2F(x,t)\right)\partial_x^3 F(x,t)}{\partial_x F(x,t)} - \frac{\left(1-2F(x,t)\right)\left(\partial_x^2 F(x,t)\right)^2}{\left(\partial_x F(x,t)\right)^2} \tag{D.2}
$$
$$
+ \partial_x \left[ \frac{\left(1-2F(x,t)\right)\partial_x^2 F(x,t)}{\partial_x F(x,t)} \right] - 2\,\partial_x \left[ \frac{F(x,t)\left(1-F(x,t)\right)\left(\partial_x^2 F(x,t)\right)^2}{\left(\partial_x F(x,t)\right)^3} \right].
$$

Next, using the algebraic identity

$$
\frac{\left(1-2F(x,t)\right)\partial_x^3 F(x,t)}{\partial_x F(x,t)} - \frac{\left(1-2F(x,t)\right)\left(\partial_x^2 F(x,t)\right)^2}{\left(\partial_x F(x,t)\right)^2} \tag{D.3}
$$
$$
= 2\,\partial_x \left(\partial_x F(x,t)\right) + \partial_x \left[ \frac{\left(1-2F(x,t)\right)\partial_x^2 F(x,t)}{\partial_x F(x,t)} \right],
$$

we express the RHS of (D.2) as a total space derivative

$$
\partial_t \rho(x,t) + \partial_x^2 \rho(x,t) = 2\,\partial_x \left[ \partial_x F(x,t) + \frac{\left(1-2F(x,t)\right)\partial_x^2 F(x,t)}{\partial_x F(x,t)} \right.
$$
$$
\left. - \frac{F(x,t)\left(1-F(x,t)\right)\left(\partial_x^2 F(x,t)\right)^2}{\left(\partial_x F(x,t)\right)^3} \right]. \tag{D.4}
$$

Then, we use a second identity derived using (24)

$$
\frac{\rho(x,t)\left(1-\rho(x,t)\right)}{F(x,t)\left(1-F(x,t)\right)} = 1 + \frac{\left(1-2F(x,t)\right)\partial_x^2 F(x,t)}{\left(\partial_x F(x,t)\right)^2}
$$
$$
- \frac{F(x,t)\left(1-F(x,t)\right)\left(\partial_x^2 F(x,t)\right)^2}{\left(\partial_x F(x,t)\right)^4}, \tag{D.5}
$$

to show that the RHS of (20a) is identical to the RHS of (D.4), thus verifying (20a).

# E   An explicit verification that the solution (23-24) satisfies the boundary condition (22)

Using (24) we write

$$
\rho - \Gamma_a\,\partial_x \rho = \left(F - \Gamma_a\,\partial_x F\right) + \frac{F\left(1-F\right)}{(\partial_x F)^2}\left(\partial_x^2 F - \Gamma_a\,\partial_x^3 F\right) - \Gamma_a\,\partial_x^2 F\,\partial_x\left[ \frac{F\left(1-F\right)}{(\partial_x F)^2} \right], \tag{E.1}
$$

At the left boundary ($x = 0$), due to the boundary condition (21a) the first parentheses on the right hand side equals to $\rho_a$. The second term vanishes due to $\partial_x^2 F = \Gamma_a \partial_x^3 F$ for $x = 0$ which is seen from

$$-(\partial_x^2 F - \Gamma_a \partial_x^3 F) = \partial_t(F - \Gamma_a \partial_x F) = 0, \tag{E.2}$$

where we used (23) and (21a). The third term

$$\Gamma_a \partial_x^2 F \partial_x \left[ \frac{F(1-F)}{(\partial_x F)^2} \right] = \Gamma_a \frac{\partial_x^2 F}{\partial_x F} \left[ 1 - 2F - \frac{2F(1-F)\partial_x^2 F}{(\partial_x F)^2} \right] = \Gamma_a \frac{(1-2\rho)(\rho - F)}{F(1-F)} \partial_x F, \tag{E.3}$$

using (24). This term further simplifies at $x = 0$ using (21a),

$$\Gamma_a \frac{(1-2\rho)(\rho - F)}{F(1-F)} \partial_x F = \frac{(1-2\rho)(\rho - F)(F - \rho_a)}{F(1-F)}. \tag{E.4}$$

Substituting these results in (E.1) we see that (22a) is satisfied. The analysis is similar for the right boundary condition (22b).

# F  Vanishing of the Hamiltonian along the optimal path

Here, we explicitly verify that the Hamiltonian (14b) vanishes for the optimal path $\widehat{\rho}(x,t)$ in (19) and $\rho(x,t)$ in (24). For this we write the bulk Hamiltonian (14c) in terms of $F(x,t)$ using (19).

$$H_{\text{bulk}} = \int_0^1 dx \left[ \frac{\rho(x,t)\big(1-\rho(x,t)\big)\big(\partial_x F(x,t)\big)^2}{\big(F(x,t)\big)^2 \big(1-F(x,t)\big)^2} - \frac{\partial_x F(x,t)\, \partial_x \rho(x,t)}{F(x,t)\big(1-F(x,t)\big)} \right], \tag{F.1}$$

The integrand in the above expression can be written as a total space derivative,

$$H_{\text{bulk}} = \int_0^1 dx\, \partial_x \left[ \frac{\big(F(x,t)-\rho(x,t)\big)\partial_x F(x,t)}{F(x,t)\big(1-F(x,t)\big)} \right], \tag{F.2}$$

This is seen by explicitly writing the spatial derivative

$$\partial_x \left[ \frac{\big(F(x,t)-\rho(x,t)\big)\partial_x F(x,t)}{F(x,t)\big(1-F(x,t)\big)} \right] = \frac{\big(\partial_x F(x,t)-\partial_x \rho(x,t)\big)\partial_x F(x,t)}{F(x,t)\big(1-F(x,t)\big)}$$
$$+ \frac{\big(F(x,t)-\rho(x,t)\big)\partial_x^2 F(x,t)}{F(x,t)\big(1-F(x,t)\big)} - \frac{\big(F(x,t)-\rho(x,t)\big)\big(1-2F(x,t)\big)\big(\partial_x F(x,t)\big)^2}{\big(F(x,t)\big)^2 \big(1-F(x,t)\big)^2}, \tag{F.3}$$

which is then simplified using an identity

$$\partial_x^2 F(x,t) = \frac{\big(\rho(x,t)-F(x,t)\big)\big(\partial_x F(x,t)\big)^2}{F(x,t)\big(1-F(x,t)\big)}, \tag{F.4}$$

that follows from (24). We get

$$\partial_x \left[ \frac{\big(F(x,t)-\rho(x,t)\big)\partial_x F(x,t)}{F(x,t)\big(1-F(x,t)\big)} \right] = \frac{\rho(x,t)\big(1-\rho(x,t)\big)\big(\partial_x F(x,t)\big)^2}{\big(F(x,t)\big)^2 \big(1-F(x,t)\big)^2}$$
$$- \frac{\partial_x F(x,t)\, \partial_x \rho(x,t)}{F(x,t)\big(1-F(x,t)\big)}, \tag{F.5}$$

where the RHS is the integrand in (F.1).

Completing the spatial integration in (F.2) gives us

$$H_{\text{bulk}} = -\frac{\big(F(0,t)-\rho(0,t)\big)\,\partial_x F(0,t)}{F(0,t)\big(1-F(0,t)\big)} + \frac{\big(F(1,t)-\rho(1,t)\big)\,\partial_x F(1,t)}{F(1,t)\big(1-F(1,t)\big)}, \tag{F.6}$$

which can be further simplified using the Robin boundary condition in (21) to

$$H_{\text{bulk}} = -\frac{\big(F(0,t)-\rho(0,t)\big)\big(F(0,t)-\rho_a\big)}{\Gamma_a\,F(0,t)\big(1-F(0,t)\big)} + \frac{\big(F(1,t)-\rho(1,t)\big)\big(\rho_b-F(1,t)\big)}{\Gamma_b\,F(1,t)\big(1-F(1,t)\big)}, \tag{F.7}$$

The two terms of $H_{\text{bulk}}$ in (F.7) precisely cancel with the boundary terms $H_{\text{left}}$ and $H_{\text{right}}$ in (14b) when expressed in terms of the $F$-field using (24). This leads to our desired result $H_{\text{bulk}} + \frac{1}{\Gamma_a} H_{\text{left}} + \frac{1}{\Gamma_b} H_{\text{right}} = 0$ along the optimal path.

# G  A derivation of the Hamilton-Jacobi equation

One way to get the Hamilton-Jacobi equation (33) is by comparing (15) with Hamilton's variational principle in classical mechanics [108]. Under a suitable canonical transformation the optimal Action $\psi$ follows

$$H\left[\frac{\delta\psi[\rho]}{\delta\rho},\rho\right] = -\partial_t\psi[\rho]. \tag{G.1}$$

For the specific problem (15) the optimal Action $\psi$ is time independent and this gives (33).

We present an alternative derivation following [47,81]. Even though our derivation may not be the simplest, we feel it presents an instructive comparison with the derivation in [47,81] for the fast coupling regime. For this comparison we rewrite (14a) in terms of a current field $j(x,t)$

$$\Pr[r(x)] = \int_{\bar\rho}^{r} [\mathcal{D}j][\mathcal{D}\rho]\left[\prod_{t,x}\delta\big(\partial_t\rho + \partial_x j\big)\right]e^{-L\int_{-\infty}^{0}\mathrm{d}t\,\mathcal{L}[j,\rho]}, \tag{G.2a}$$

where the Action

$$\mathcal{L} = \int_0^1 \mathrm{d}x\,\frac{\big(j(x,t)+\partial_x\rho(x,t)\big)^2}{4\rho(x,t)(1-\rho(x,t))} + \Phi_{\text{lft}}\big(j(0,t),\rho(0,t)\big) + \Phi_{\text{rgt}}\big(j(1,t),\rho(1,t)\big), \tag{G.2b}$$

with

$$\Phi_{\text{lft}}(j,\rho) = \min_{\widehat\rho}\left(\widehat\rho\,j - \frac{\omega(\widehat\rho,\rho_a,\rho)}{\Gamma_a}\right), \quad \Phi_{\text{rgt}}(j,\rho) = \min_{\widehat\rho}\left(\widehat\rho\,j - \frac{\omega(\widehat\rho,\rho,\rho_b)}{\Gamma_b}\right). \tag{G.2c}$$

One way to verify this representation is by proving (14) starting from (G.2) following these algebraic steps in order: (1) introduce $\widehat\rho$ by an integral representation of the delta function. (2) Perform an integration by parts for the term $\widehat\rho\,\partial_x j$ so that the terms involving $j$ can be written as full square. (3) Do the Gaussian integration over the $j$-field. (4) Considering large $L$, for the $\Phi_{\text{lft}}$ in (G.2c) replace the minimum of the function by the function itself and similarly for the $\Phi_{\text{rgt}}$. This would cancel the boundary term $\widehat\rho(0,t)\,j(0,t)$ (and similarly on the right boundary).

For deriving (G.1) we consider a transition probability $\Pr_{(-\mathrm{d}t)}[s(x)]$, analogous to (G.2a), starting with the same initial profile $\rho(x,-\infty) = \bar\rho(x)$ but ending at time $(0-\mathrm{d}t)$ at density $s(x)$. The two probabilities are related

$$\Pr[r(x)] = \int [\mathcal{D}s]\,\mathcal{P}(r,s,\mathrm{d}t)\,\Pr_{(-\mathrm{d}t)}[s(x)], \tag{G.3}$$

where $\mathcal{P}(r, s, \mathrm{d}t)$ is the transition probability between density profiles $s(x)$ and $r(x)$ at time $\mathrm{d}t$ apart. For infinitesimal $\mathrm{d}t$, from (G.2a)

$$\mathcal{P}(r, s, \mathrm{d}t) \simeq \int [\mathcal{D}j] \left[ \prod_x \delta\big(r(x) - s(x) + \mathrm{d}t \, \partial_x j\big) \right] e^{-L \, \mathrm{d}t \, \mathcal{L}[r, j]}. \tag{G.4}$$

For large $L$, considering a large deviation asymptotic $\mathrm{Pr}_{(-\mathrm{d}t)}[s(x)] \sim e^{-L\psi_{-\mathrm{d}t}[s(x)]}$ along with (4a), we get from (G.3)

$$\psi[r(x)] \simeq \min_{j(x)} \big( \psi_{-\mathrm{d}t}[r(x) + \mathrm{d}t \partial_x j] + \mathrm{d}t \, \mathcal{L}[r(x), j(x)] \big). \tag{G.5}$$

A Taylor series expansion for small $\mathrm{d}t$ gives

$$\partial_t \psi[r(x)] = \min_{j(x)} \left( \int_0^1 \mathrm{d}x \, \frac{\delta \psi[r(x)]}{\delta r(x)} \partial_x j(x) + \mathcal{L}[r(x), j(x)] \right). \tag{G.6}$$

The minimization is done using an integration by parts, which leads to

$$\partial_t \psi[r(x)] = \min_{j(x)} \left\{ \int_0^1 \mathrm{d}x \left[ \frac{\big(j(x) + r'(x)\big)^2}{4 \, r(x)\big(1 - r(x)\big)} - j(x) \, \partial_x \frac{\delta \psi[r(x)]}{\delta r(x)} \right] \right.$$
$$\left. + \left( \Phi_{\mathrm{lft}}\big(j(0), r(0)\big) - j(0) \frac{\delta \psi[r(0)]}{\delta r(0)} \right) + \left( \Phi_{\mathrm{rgt}}\big(j(1), r(1)\big) + j(1) \frac{\delta \psi[r(1)]}{\delta r(1)} \right) \right\}, \tag{G.7}$$

where we explicitly wrote (G.2b). Minimization of the quadratic term in $j(x)$ for $0 < x < 1$ gives the bulk Hamiltonian term in (G.1) whereas minima of the boundary terms are the boundary Hamiltonians due to the inverse of the transformation (G.2c). Considering that $\psi$ is independent of time we get the Hamilton-Jacobi equation (33).

   **Remark**: In the derivation [47, 81] for the fast coupling regime, the boundary terms are discarded by imposing a vanishing $\frac{\delta \psi[r]}{\delta r}$ at the boundary that relates to the Dirichlet condition on density. The expression in (G.7) presents a natural explanation why it is so in the fast coupling limit.

## H   A solution of the Hamilton-Jacobi equation

A verification for the solution (4b) of the Hamilton-Jacobi equation (33) is effectively given already in Appendix F. We reiterate the algebraic steps for completeness, which are very similar to the discussion in [47] for the fast coupling limit.

   For (4b),

$$\frac{\delta \psi[r(x)]}{\delta r(x)} = \log \frac{r(x)\big(1 - F(x)\big)}{F(x)\big(1 - r(x)\big)}, \tag{H.1}$$

and this gives

$$H_{\mathrm{bulk}}\left[ \frac{\delta \psi[r(x)]}{\delta r(x)}, r(x) \right] = \int_0^1 \mathrm{d}x \left[ \frac{r(x)\big(1 - r(x)\big)\big(F'(x)\big)^2}{\big(F(x)\big)^2\big(1 - F(x)\big)^2} - \frac{r'(x) F'(x)}{F(x)\big(1 - F(x)\big)} \right], \tag{H.2}$$

for the bulk Hamiltonian (14c) and similarly for the boundary Hamiltonians

$$H_{\mathrm{left}}\left[ \frac{\delta \psi[r(x)]}{\delta r(0)}, r(0) \right] = \frac{\big(F(0) - r(0)\big)\big(F(0) - \rho_a\big)}{F(0)\big(1 - F(0)\big)}, \tag{H.3}$$

$$H_{\mathrm{right}}\left[ \frac{\delta \psi[r(x)]}{\delta r(1)}, r(1) \right] = \frac{\big(F(1) - r(1)\big)\big(F(1) - \rho_b\big)}{F(1)\big(1 - F(1)\big)}. \tag{H.4}$$

Following an algebra similar to what was discussed in Appendix F, we verify that the total Hamiltonian vanishes, thus satisfying (33).

# I  Slow coupling limit of the density ldf

To keep our discussion simple, we consider $\Gamma_a = \Gamma_b \equiv \Gamma$ and analyze the slow coupling limit $\Gamma \to \infty$ of the ldf (4b). For small $\Gamma$, we write a series expansion $F = F_0 + \frac{1}{\Gamma} F_1 + \frac{1}{\Gamma^2} F_2 + \cdots$ where comparing with equilibrium-ldf [1] we assume $F_0$ to be a constant, independent of $x$. With this assumption we show that the slow coupling result (2b) is a limit of the ldf (4b).

Using the expansion of $F(x)$ in the spatial boundary condition (4c) we get

$$\rho_a = \left(F_0 - F_1'(0)\right) + \frac{1}{\Gamma}\left(F_1(0) - F_2'(0)\right) + \cdots, \tag{I.1a}$$

$$\rho_b = \left(F_0 + F_1'(1)\right) + \frac{1}{\Gamma}\left(F_1(1) + F_2'(1)\right) + \cdots, \tag{I.1b}$$

and similarly from the relation between $\rho(x)$ and $F(x)$ in (1c), we obtain

$$\rho(x) = \Gamma \frac{F_0 (1 - F_0) F_1''(x)}{F_1'(x)^2} + F_0 + \left[\frac{(1 - 2F_0) F_1(x) F_1'(x) - 2 F_0 (1 - F_0) F_2'(x)}{F_1'(x)^3}\right] F_1''(x)$$

$$+ \frac{F_0 (1 - F_0) F_2''(x)}{F_1'(x)^2} + \mathcal{O}\left(\Gamma^{-1}\right). \tag{I.2}$$

The left hand side of (I.2) is independent of $\Gamma$, which demands $F_1''(x) = 0$ for all $x$, simplifying the relation to

$$\rho(x) = F_0 + \frac{F_0 (1 - F_0) F_2''(x)}{F_1'(x)^2}. \tag{I.3}$$

Similarly from (I.1), we get $F_1'(0) = F_0 - \rho_a$ and $F_1'(1) = \rho_b - F_0$, which combined with $F_1''(x) = 0$ and constant $F_0$ results in

$$F_0 = \bar{\varrho} = \frac{1}{2}(\rho_a + \rho_b), \tag{I.4a}$$

$$F_1(x) = \frac{\rho_b - \rho_a}{2}\left(x - \frac{1}{2}\right). \tag{I.4b}$$

The constant term in $F_1(x)$ is determined using the average stationary state density profile (32) expanded in $\Gamma$ as

$$\bar{\rho}(x) = \bar{\varrho} - \frac{\rho_a - \rho_b}{2\Gamma}\left(x - \frac{1}{2}\right) + \frac{\rho_a - \rho_b}{4\Gamma^2}\left(x - \frac{1}{2}\right) + \cdots, \tag{I.5}$$

for which the ldf vanishes and $F(x) = \bar{\rho}(x)$.

These results for $F_0$ and $F_1(x)$ along with the spatial boundary conditions $F_2'(0) = F_1(0)$ and $F_2'(1) = -F_1(1)$ imposes an important condition for the density in (I.3):

$$\int_0^1 dx\, \rho(x) = \bar{\varrho}. \tag{I.6}$$

With the help of the expansion for $F(x)$ with (I.4) the ldf for the marginal boundary coupling (4b) to the leading order in large $\Gamma$ is

$$\psi\left[\rho(x)\right] = \int_0^1 dx\left[\rho(x)\log\frac{\rho(x)}{1 - \rho(x)} - \rho(x)\log\frac{\bar{\varrho}}{1 - \bar{\varrho}} + \log\frac{1 - \rho(x)}{1 - \bar{\varrho}} + (1 + 2\Gamma)\log\frac{1}{2\Gamma}\right].$$

The expression further simplifies using (I.6) yielding

$$\psi\left[\rho(x)\right] = \int_0^1 dx \left[ \rho(x)\log\frac{\rho(x)}{1-\rho(x)} - \bar{\varrho}\log\frac{\bar{\varrho}}{1-\bar{\varrho}} + \log\frac{1-\rho(x)}{1-\bar{\varrho}} \right] + (1+2\Gamma)\log\frac{1}{2\Gamma} \,. \quad \text{(I.7)}$$

The integral term is the slow-coupling-ldf (2b) with $\rho = \bar{\varrho}$. The additional constant term, diverging for $\Gamma \to \infty$, represents fluctuations of $\varrho$ in (2c).

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
