# Peer review of "Large Deviations in the Symmetric Simple Exclusion Process with Slow Boundaries: A Hydrodynamic Perspective"

_SciPost Physics, doi:SciPost Phys. 17, 033 (2024)_

## Round 2 · Referee Report · Anonymous (Referee 1) · 2024-7-5

Strengths

In the context of the symmetric simple exclusion process with open boundaries, the hydrodynamic approach of macroscopic fluctuation is employed systematically to rederive fluctuation results obtained by earlier more
model specific methods. The detailed computations are explained in the appendices, which thereby serve as a blueprint for other models.

Weaknesses

One might argue that there are too little novel results. But in my judgement this is balanced by demonstrating a method which has the potential of being applied
to other models, for which it is unlikely to have such exact identities as for the SSEP.

Report

The article is very well and carefully written. It is accessible to a larger readership. At first sight the long list of appendices is somewhat unusual. But at second reading, such separation has the advantage that the necessary computations are being spelled out in detail.

The article amply meets the level of SciPost and I recommend publication in its present form.

Requested changes

The notion "marginal coupling" is not optimal. There is a risk of confusion.
The term "marginal boundary coupling" would be already much better.

Recommendation

Publish (easily meets expectations and criteria for this Journal; among top 50%)

  • validity: top
  • significance: high
  • originality: top
  • clarity: top
  • formatting: excellent
  • grammar: excellent

Author:  Soumyabrata Saha  on 2024-07-16  [id 4624]

(in reply to Report 1 on 2024-07-05)
Category:
remark

We sincerely thank the referee for their valuable comments. We would like to address the concerns raised by the referee regarding a perceived minor weakness in our work. Some of the results derived in our current submission have been previously treated in the existing literature using different model-specific microscopic methods, and there is concern about "too few novel results."

It is indeed true that the final expressions for the large deviation functions (LDF) of density and current were derived using microscopic techniques by Derrida, Hirschberg, and Sadhu in 2019 [48]. However, what was not previously understood is how the rare fluctuations are generated, specifically the optimal paths of evolution that give rise to these observed fluctuations for density and for current.

Earlier works by Bertini et al. [39] using the Macroscopic Fluctuation Theory (MFT) verified the LDF of density in the limiting case of fast boundary coupling by addressing the Hamilton-Jacobi equations, which again do not provide information about the optimal paths of evolution.

Our work achieves these by an exact solution of the Euler-Lagrange equations using a novel local transformation. This contrasts with a two-step non-local transformation used in [40,41] for a similar solution, but only in the fast coupling limit. Furthermore, our solution method is generalisable to a broader class of model systems beyond the Symmetric Simple Exclusion Process (SSEP). This, in our understanding, is an important advancement considering recent attention to the integrability [18] of the Euler-Lagrange equations for SSEP, where an explicit solution at arbitrary times remains challenging.

From a technical point of view, the solution for the marginal (and slow) boundary coupling boundary condition is non-trivial. It also addresses how the boundary conditions in the fast coupling limit naturally emerge, which were previously argued heuristically. These novelties have already been emphasised in the third paragraph on page three of our Submission.

Lastly, our derivation of the MFT action is independent and does not assume any a priori conditions imposed on the system. This provides a more robust technique that can be applied to a variety of models, extending beyond the SSEP.

---

## Round 2 · Referee Report · Anonymous (Referee 2) · 2024-7-10

Report

Outside equilibrium, the steady state of a physical system and its fluctuations are sensitive to the boundary conditions. A natural question arises: how sensitive or robust are the fluctuations, particularly the associated large deviation functions, to the details of the boundary?
In the manuscript currently under review, the authors address this problem for the Symmetric Exclusion Process (SEP). Specifically, they apply Macroscopic Fluctuation Theory (MFT) to compute the large deviation functions for density and current in the SEP with slow boundaries.
While these quantities had been previously computed using the exact solution of the microscopic dynamics, it is quite important to be able to recover these results using an approach like MFT.
This approach has the potential to be broadly applicable, particularly to models that are not exactly solvable at the microscopic level. Therefore, it is relevant to understand, as done in the paper, how specific boundary conditions at the microscopic level are reflected in the MFT
formalism.
The paper is carefully written, and has been a pleasure to read. It contains original results that deserve publication.

Recommendation

Publish (easily meets expectations and criteria for this Journal; among top 50%)

  • validity: top
  • significance: high
  • originality: high
  • clarity: high
  • formatting: excellent
  • grammar: excellent

Author:  Soumyabrata Saha  on 2024-07-16  [id 4626]

(in reply to Report 2 on 2024-07-10)
Category:
remark

We sincerely thank the referee for their valuable comments regarding our Submission.

---

## Round 2 · Referee Report · Anonymous (Referee 3) · 2024-7-15

Strengths

This article presents in a coherent and clean way the derivation of large deviations from the macroscopic fluctuation functional

Weaknesses

The weakness is that some of the points have been treated in previous literature, for which references are carefully given

Report

The criteria are easily met

Recommendation

Publish (easily meets expectations and criteria for this Journal; among top 50%)

  • validity: top
  • significance: high
  • originality: good
  • clarity: top
  • formatting: perfect
  • grammar: perfect

Author:  Soumyabrata Saha  on 2024-07-16  [id 4625]

(in reply to Report 3 on 2024-07-15)
Category:
remark

We sincerely thank the referee for their valuable comments. We would like to address the concerns raised by the referee regarding a perceived weakness in our work. Some of the results derived in our current submission have been previously treated in the existing literature using different model-specific microscopic methods.

It is indeed true that the final expressions for the large deviation functions (LDF) of density and current were derived using microscopic techniques by Derrida, Hirschberg, and Sadhu in 2019 [48]. However, what was not previously understood is how the rare fluctuations are generated, specifically the optimal paths of evolution that give rise to these observed fluctuations for density.

Earlier works by Bertini et al. [39] using the Macroscopic Fluctuation Theory (MFT) verified the LDF of density in the limiting case of fast boundary coupling by addressing the Hamilton-Jacobi equations, which do not provide information about the optimal paths of evolution.

Our work achieves this by an exact solution of the Euler-Lagrange equations using a novel local transformation. This contrasts with a two-step non-local transformation used in [40,41] for a similar solution, but only in the fast coupling limit. Furthermore, our solution method is generalisable to a broader class of model systems beyond the Symmetric Simple Exclusion Process (SSEP). This, in our understanding, is an important advancement considering recent attention to the integrability [18] of the Euler-Lagrange equations for SSEP, where an explicit solution at arbitrary times remains challenging.

From a technical point of view, the solution for the weak coupling boundary condition is non-trivial. It also addresses how the boundary conditions in the fast coupling limit naturally emerge, which were previously argued heuristically. These novelties have already been emphasised in the third paragraph on page three of our submission.

Lastly, our derivation of the MFT action is independent and does not assume any a priori conditions imposed on the system. This provides a more robust technique that can be applied to a variety of models, extending beyond the SSEP.

---

## Editorial Decision

published